# A high-throughput, flow cytometry approach to measure phase behavior and exchange in biomolecular condensates

Yuchen He[1,6], George M. Ongwae[1,6], Anupam Mondal [2], Joel A. Moses [1], Jeetain Mittal [2,3,4] & Marcos M. Pires [1,5] ✉

Biomolecular condensates are essential for cellular organization, yet their formation dynamics and molecular content exchange properties remain poorly understood. Here we show that flow cytometry provides a high-throughput, solution-based platform for analyzing condensate behavior at the single-droplet level. Using self-interacting NPM1 condensates as a model, we demonstrate that this approach quantifies phase behavior across protein and salt conditions, measures the partitioning of diverse macromolecules—including antibodies, lipids, small-molecule drugs, and RNA—and detects molecular colocalization with high statistical precision. Importantly, we establish a high-throughput assay to track real-time molecular exchange between preformed condensates and newly added, orthogonally tagged protein. These measurements reveal that condensate aging significantly reduces molecular dynamisms, likely due to altered biophysical properties with time. Compared to conventional imaging techniques that require surface immobilization or complex instrumentation, our method enables rapid, quantitative characterization of condensate dynamics and molecular content, providing a scalable framework for probing condensate function.

Biomolecular condensates represent a fundamental organizing principle in cell biology, enabling the compartmentalization and regulation of biochemical reactions without the need for membrane-bound organelles[1,2]. These dynamic structures form via liquid-liquid phase separation (LLPS), a physicochemical process in which biomolecules demix from the surrounding milieu to generate dense, droplet-like assemblies[3,4]. LLPS is driven by multivalent interactions among proteins and nucleic acids, often facilitated by intrinsically disordered regions (IDRs), low-complexity domains, and specific modular interaction motifs[5–8]. By concentrating proteins, nucleic acids, and other macromolecules, condensates orchestrate important biochemical and biophysical processes such as gene regulation, signal transduction, RNA metabolism, and stress response (Fig. 1a)[9–11]. Their ability to

assemble and disassemble in response to environmental cues is essential for cellular homeostasis and maintenance. Disruptions in condensate formation or changes in their material properties have been implicated in various pathologies including amyotrophic lateral sclerosis (ALS), frontotemporal dementia, and several cancers[12–16]. Understanding the molecular principles governing condensate assembly, composition, and dynamics is therefore critical for elucidating cellular functional organization and developing targeted therapies.

As our understanding of condensate biology expands, increasing attention has turned to how specific partner molecules interact with or partition into these structures[17–19]. While initial studies largely examined the roles of proteins and nucleic acids in driving or enriching

[1]Department of Chemistry, University of Virginia, Charlottesville, VA, USA. [2]Artie McFerrin Department of Chemical Engineering, Texas A&M University, College Station, TX, USA. [3]Department of Chemistry, Texas A&M University, College Station, TX, USA. [4]Interdisciplinary Graduate Program in Genetics and Genomics, Texas A&M University, College Station, TX, USA. [5]Department of Microbiology, Immunology, and Cancer, University of Virginia, Charlottesville, VA, USA. [6]These authors contributed equally: Yuchen He, George M. Ongwae. ✉e-mail: mpires@virginia.edu

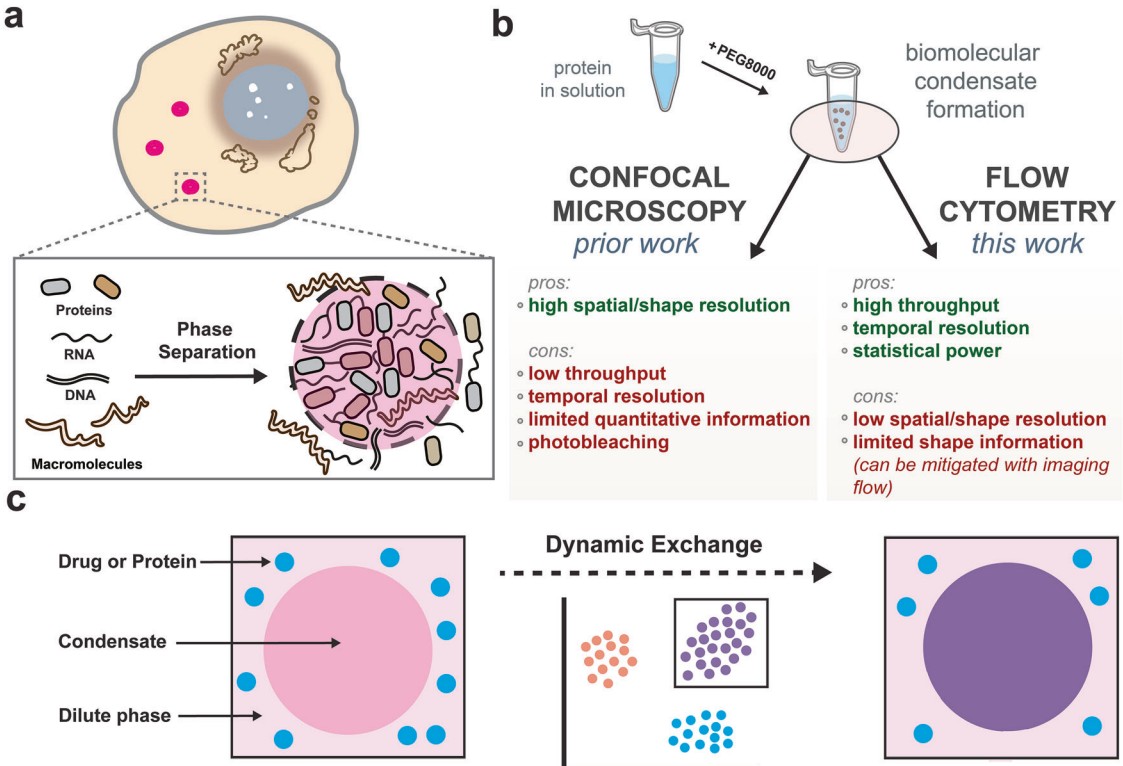

**Fig. 1 | Developing flow cytometry-based strategies to study biomolecular condensates. a** Schematic illustration of biomolecular condensate formation via liquid-liquid phase separation (LLPS) from proteins, RNAs, and DNAs in cells. **b** Comparison of confocal microscopy and flow cytometry for condensate analysis. **c** Conceptual overview showing dynamic exchange between condensates and surrounding molecules. Flow cytometry enables continuous tracking of such exchange events over time, providing a powerful approach to assess condensate dynamics.

within condensates, recent work has broadened this view to include small molecules. For example, one early discovery revealed that cyclic GMP-AMP synthase can undergo phase separation upon binding to oligonucleotides, with these nucleic acids enriching within the resulting condensates[20]. Building on this, researchers have begun to systematically investigate which classes of molecules can diffuse into, concentrate within, or be excluded from existing condensates. These efforts have primarily focused on small molecules, particularly in the context of how condensates might influence drug distribution. Notably, the Young lab demonstrated that fluorophore-tagged FDA-approved therapeutics showed differential localization across diverse protein condensates[21]. This work has since expanded to BODIPY-based libraries to further explore physicochemical determinants of condensate partitioning[22]. More recently, a wide scan of over 1000 small-molecule drugs was evaluated for their colocalization with condensates[23].

Despite growing recognition of their significance, in vitro analyses of biomolecular condensates remain challenging tasks, largely constrained by methodological limitations. Confocal microscopy has long been an often-used method to analyze condensate formation, dynamics, and composition, offering high spatial resolution[20–22]. However, its low throughput in most settings limits experimental scalability, and the temporal resolution is often insufficient to capture rapid dynamics. Moreover, manual image processing can also introduce user bias. Other commonly used methods, such as fluorescence recovery after photobleaching (FRAP), turbidity measurement, and sedimentation assays[24–26], provide insights into molecular mobility or condensate composition but are either low throughput or lack spatial and temporal resolution. These techniques often fail to capture condensate heterogeneity or rapid dynamic changes in a quantitative manner. While alternative tools such as microfluidic platforms are

being explored[27], there remains a need for robust, quantitative, and scalable methods to study condensate behavior with widely accessible instrumentation. In parallel, emerging cell-based approaches such as optogenetic systems enable controlled condensate assembly and live-cell imaging platforms that track condensate dynamics[28,29]. However, these approaches remain technically demanding and are not inherently high-throughput, limiting their scalability for systematic screening.

To address these limitations, we provide evidence that flow cytometry can be a complementary method for condensate analysis, enabling high-throughput and quantitative information without the constraints of surface immobilization or limited imaging volumes (Fig. 1b). Our results show that we can specifically measure condensate size, morphology, and fluorescence intensity as key readouts of condensate formation, composition, and dissolution. Conventional flow cytometry allows assessment of thousands to millions of individual condensates within minutes, providing high-level statistical power to provide broad descriptions across droplet populations. Compared to confocal microscopy, flow cytometry mostly eliminates issues associated with photobleaching, minimizes user bias, and delivers reproducible fluorescence measurements under consistent excitation conditions. Confocal imaging retains complementary advantages in spatial resolution and direct visualization of condensate morphology, and other commonly used assays (e.g., FRAP) can provide high spatial resolution on molecular mobility.

To benchmark the potential of flow cytometry for characterizing biomolecular condensates, we first conducted a series of experiments comparing results obtained by confocal microscopy with those from flow cytometry. Our data show that flow cytometry reliably captures condensate formation, composition, and dissolution, closely aligning with imaging-based observations. Importantly, this technique also

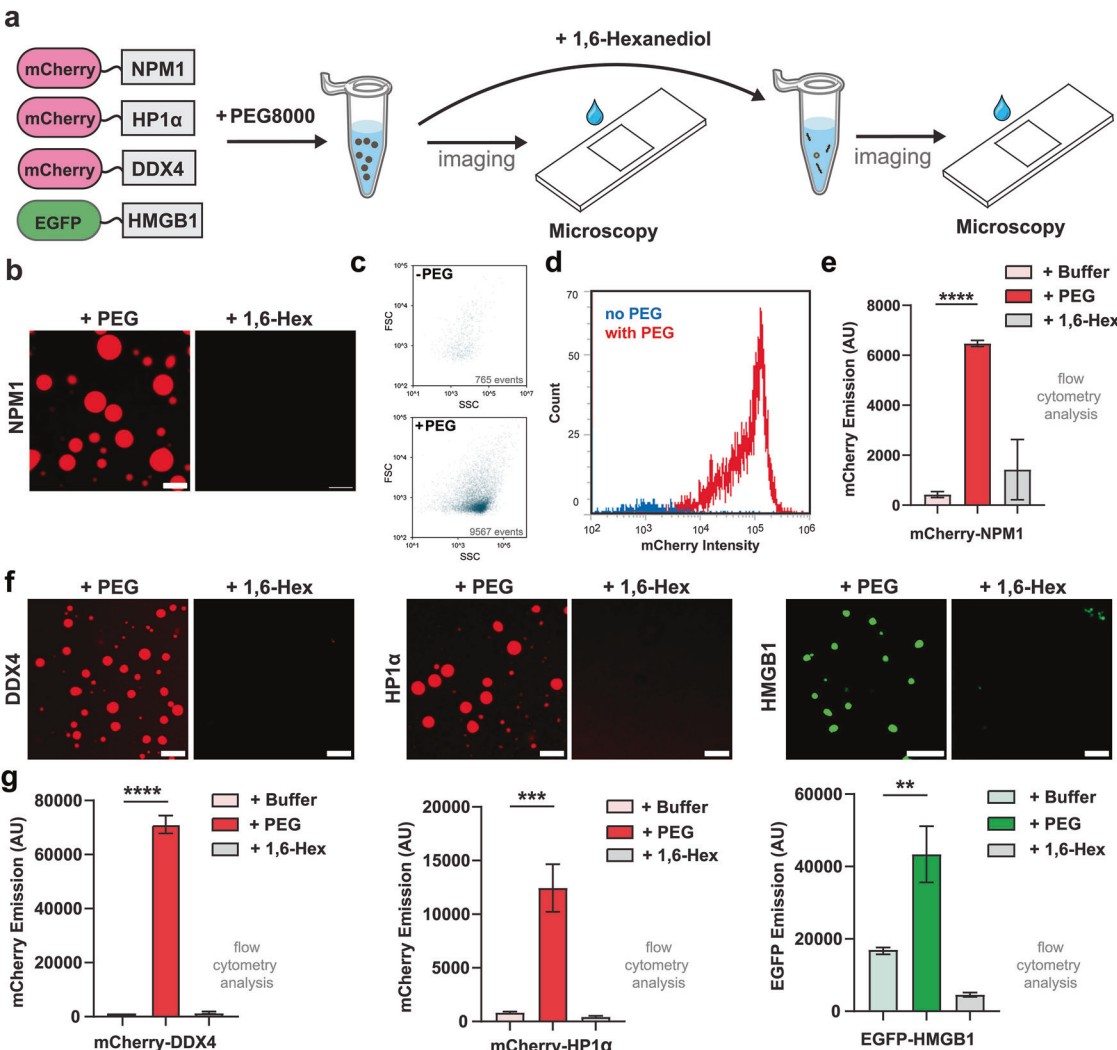

**Fig. 2 | Characterization of condensate formation and dissolution using confocal microscopy and flow cytometry. a** Schematic overview of the in vitro droplet assay workflow. Purified fluorescently tagged proteins (mCherry-NPM1, mCherry-DDX4, mCherry-HP1α, and EGFP-HMGB1) were incubated with 10% PEG8000 to induce condensate formation, followed by treatment with 10% 1,6-hexanediol to assess reversibility. **b** Confocal images and flow cytometry analysis of 20 μM mCherry-NPM1 under control, PEG, and PEG + hexanediol conditions. PEG treatment induces bright, spherical condensates, which are disrupted by 1,6-hexanediol. Flow cytometry confirms increased fluorescence intensity with PEG and reduction upon hexanediol treatment, indicating reversible condensate formation. **c** Forward scatter (FSC) versus side scatter (SSC) plots of mCherry-NPM1 with and without PEG. Under equal acquisition time (30 s), PEG treatment resulted in a markedly larger number of detected events (9567 vs. 765). **d** Histogram of mCherry

fluorescence intensity for mCherry-NPM1 with and without PEG. PEG treatment caused a clear rightward population shift, reflecting enhanced condensate-associated signal. **e** Quantification of mean fluorescence intensity (MFI) of mCherry-NPM1 under no PEG, with PEG, and PEG + 1,6-hexanediol conditions, showing reversible condensate formation and dissolution. **f** Confocal microscopy images of mCherry-DDX4, mCherry-HP1α, and EGFP-HMGB1 under no PEG, with PEG, and PEG + 1,6-hexanediol conditions. **g** Flow cytometry MFI analyses of mCherry-DDX4, mCherry-HP1α, and EGFP-HMGB1 under the same conditions. Flow cytometry data represent properly gated populations of fluorescent particles. Statistical analysis was performed using unpaired, two-tailed student's *t*-tests. Error bars indicate standard deviation (SD). Significance is denoted as follows: $p < 0.05$ (*), $p < 0.01$ (**), $p < 0.001$ (***), and $p < 0.0001$ (****). Scale bars = 5 μm. Exact p-values are provided in the Source Data file.

enables real-time monitoring of protein exchange between condensates and their surroundings (Fig. 1c), offering new kinetic insights into phase-separated systems. To further dissect the molecular basis of condensate formation, we complemented our experimental work with coarse-grained molecular dynamics simulations. In parallel, to mechanistically understand the exchange dynamics during aging, we developed a discrete-state stochastic model that quantitatively supports our experimental findings using first-passage calculations. By expanding the analytical toolkit for condensate research, this study establishes flow cytometry as a robust and scalable approach for probing phase-separated systems. Its capacity to resolve dynamic behaviors with high temporal and statistical precision opens new avenues for mechanistic insight and may inform therapeutic strategies targeting condensate-associated pathologies.

## Results and discussion

### Characterization of condensate formation and dissolution

Unlike cellular condensates that assemble under physiological conditions, in vitro condensate formation typically requires crowding agents to drive phase separation. Reagents such as polyethylene glycol (PEG) and Ficoll mimic the intracellular macromolecular environment by increasing effective protein concentration and promoting demixing through excluded volume effects[30,31]. As an initial test, we selected four well-characterized scaffolding proteins—NPM1, HP1α, DDX4, and HMGB1—each previously reported to undergo phase separation and function in chromatin organization, RNA metabolism, and nucleolar assembly[32–36]. Each protein was fused to a fluorescent tag to facilitate visualization of condensate behavior (Fig. 2a). The recombinant proteins were purified and subjected to in vitro droplet assays to assess

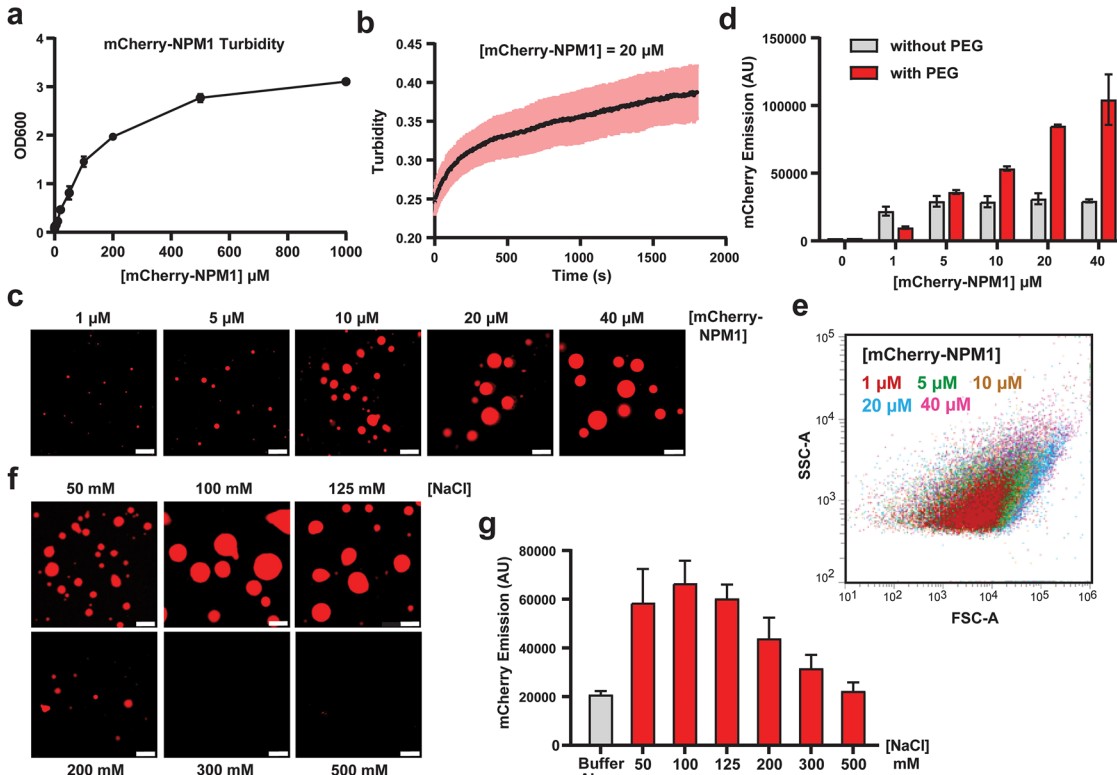

**Fig. 3 | Characterization of NPM1 condensates using a combination of diverse approaches. a** Turbidity measurements at $OD_{600}$ show a concentration-dependent increase, indicating higher condensate formation at increasing NPM1 concentrations. **b** Time-course turbidity measurements of 20 μM NPM1 reveal rapid condensate formation. **c** Confocal images of mCherry-NPM1 droplets at different protein concentrations show larger droplet sizes with increasing concentration. **d** Flow cytometry analysis is consistent with **c**, showing that higher mCherry-NPM1 concentrations result in increased fluorescence intensity, reflecting more

condensate formation. **e** Forward scatter (FSC-A) and side scatter (SSC-A) analysis of mCherry-NPM1 condensates indicate that larger condensates exhibit higher light scattering, consistent with increased droplet size. **f** Confocal images of 20 μM mCherry-NPM1 droplets in buffers with varying NaCl concentrations show that condensates remain stable below 125 mM NaCl but dissolve at higher salt concentrations (>300 mM). **g** Flow cytometry analysis aligns with the salt titration data in **f**, confirming the salt-dependent dissolution of NPM1 condensates. Scale bars = 5 μm.

condensate formation. Specifically, each protein was incubated with 10% (w/v) PEG for 30 min and initially imaged by confocal microscopy using the workflow outlined in Fig. 2a. As expected, each protein formed spherical condensates in the presence of PEG, confirming their phase separation propensity (Fig. 2b, f). To assess the sensitivity of these condensates to disruption, we also treated the assembled droplets with 1,6-hexanediol, a known disruptor of biomolecular condensates[37,38]. The addition of 1,6-hexanediol led to the dissolution of these droplets, demonstrating the reversibility of condensate formation.

Next, we set out to validate flow cytometry as a method to analyze NPM1-based droplets. To validate the confocal observations using flow cytometry, we analyzed the same samples in parallel using both techniques. First, forward scatter (FSC) and side scatter (SSC) plots were compared for mCherry-NPM1 in the absence and presence of PEG (Fig. 2c). In the absence of PEG treatment, few events were detected. Under the same acquisition time window, the addition of PEG to the same NPM1 solution led to a significant increase in registered events, consistent with the induction of droplet formation. Because light scattering depends not only on size but also on refractive index and internal structure, FSC and SSC should be regarded as optical proxies rather than direct measures of absolute dimensions[39]. Next, histogram analysis of mCherry fluorescence intensity further corroborated a clear increase in the PEG-treated sample relative to the untreated control, a finding that is in agreement with the increased level of protein per event in droplet formation (Fig. 2d). Finally, mean fluorescence intensity (MFI) measurements showed a marked increase with PEG and a significant decrease after 1,6-hexanediol treatment, confirming both

the formation and dissolution of condensates (Fig. 2e). Similar trends were observed for DDX4, HP1α, and HMGB1, with microscopy and cytometry providing complementary evidence (Fig. 2f, g). Histogram figures and scatter/fluorescence distribution plots for the other proteins are shown in Supplementary Fig. 1. Collectively, these data demonstrate that flow cytometry robustly captures condensate formation and disruption with statistical power across tens of thousands of events, while microscopy provides high-resolution spatial context. We envision that this modality of analysis could be paired with large-scale small molecule libraries to discover novel drug-like agents that can block formation or disrupt droplets.

## Characterization of NPM1 condensates using experimental and computational approaches

To gain deeper insight into the condensate behavior of NPM1, we conducted a series of downstream characterization experiments. Turbidity measurements of mCherry-NPM1 in the presence of PEG revealed a concentration-dependent increase in light scattering, consistent with enhanced condensate formation (Fig. 3a). Real-time monitoring of NPM1 turbidity following PEG addition showed a progressive, time-dependent increase, consistent with prior observations of condensate (droplet) formation (Fig. 3b). Confocal microscopy revealed that increasing protein concentrations led to the formation of a higher density and visibly larger droplets (Fig. 3c), reflecting enhanced phase separation under higher protein concentrations. Critically, flow cytometry revealed a clear concentration dependence, showing a progressive increase in mCherry fluorescence intensity with increasing protein concentration (Fig. 3d). Quantitatively, flow

cytometry–derived median fluorescence measured on the same samples correlated strongly with confocal image medians (Pearson's $r = 0.97$, $p < 0.01$; Supplementary Fig. 2), indicating that these orthogonal assays report consistent concentration-dependent trends while providing complementary readouts: confocal imaging resolves droplet morphology and spatial distribution, turbidity reports bulk light scattering, and flow cytometry enables rapid, population-level quantification across tens of thousands of events. In addition, forward and side scatter plots (FSC vs. SSC) indicated a size-dependent shift in condensate populations, supporting the expected increase in the size of the formed droplets at higher protein concentrations (Fig. 3e). While larger condensates generally exhibited higher scattering signals, the distributions did not follow a strictly linear trend, reflecting the previously observed heterogeneity in condensate morphology, internal structure, and refractive index. This non-linear relationship is consistent with predictions from Mie scattering theory, where scattering intensity depends on both particle size and refractive index contrast in a complex manner rather than scaling linearly with size[40,41]. Condensate diameter analysis from confocal images (Supplementary Fig. 2d) further confirmed the heterogeneity of condensate sizes, supporting the notion that both small and large droplets contribute to the observed bulk and flow cytometry signals.

Recognizing that many condensate-driving interactions are electrostatic in nature[42,43], we next evaluated whether flow cytometry could accurately capture the expected salt-dependent (NaCl) changes in droplet behavior described in prior studies[44]. Upon titration of NaCl from 50 to 500 mM, a clear bimodal response in droplet size was observed, characterized by an initial increase at lower salt concentrations followed by disruption of droplet formation at higher salt levels (Fig. 3f). This observation is consistent with electrostatic screening effects at higher concentrations, where increased ionic strength weakens multivalent protein-protein and protein-RNA interactions essential for condensate stability[45]. Importantly, flow cytometry results mirrored this trend with corresponding decreases in fluorescence intensity at higher salt concentrations (Fig. 3g).

An important consideration for flow cytometry is whether the focused fluidics generate shear stress or whether the buffer conditions compromise condensate integrity. Previous work has shown that active transport and dynamic cellular processes can reinforce condensate stability, suggesting that physical stress does not necessarily disrupt their integrity[46]. To confirm this, we analyzed mCherry-NPM1 condensates using imaging flow cytometry (IFC), which combines the high-throughput, quantitative capabilities of flow cytometry with high-resolution imaging across multiple channels[47,48]. While typically used for cellular analysis, IFC has been applied to study protein aggregates and extracellular vesicles[49,50], demonstrating its utility for analyzing non-cell-based biomaterials. IFC confirmed that increasing protein concentrations resulted in larger condensates with higher levels of fluorescence in both bright field and fluorescence channels (Supplementary Fig. 3a). Plotting surface area (bright field) against fluorescence intensity (mCherry) demonstrated a concentration-dependent shift toward larger-sized events (Supplementary Fig. 3b). Quantification revealed a concentration-dependent increase in condensate size and fluorescence intensity (Fig. 4a), consistent with confocal microscopy results (Fig. 3c). The mean fluorescence intensity measured by IFC also closely matched the data obtained from standard flow cytometry (Fig. 3d, additional representative images are shown in Supplementary Fig. 4). Critically, these results provided key evidence that the events registered in a standard laboratory flow cytometer mirror the general dimensional features of the condensates observed using a high-end technique that combines both imaging and the throughput of flow cytometry.

## Molecular dynamics studies of NPM1

While experimental approaches provide crucial insight into the morphology of NPM1 condensates, they do not directly reveal how intramolecular architecture contributes to phase separation. Specifically, it remains unclear how domain-specific interactions within NPM1 drive condensate formation and what role individual NPM1 domains play in self-association. Previous studies have largely focused on how NPM1 interacts with RNA or other macromolecular partners, leaving gaps in our understanding of how NPM1 alone can organize into phase-separated assemblies[51–53]. To address this, we performed coarse-grained (CG) molecular dynamics simulations to investigate how structural domains within NPM1 contribute to self-association and condensate formation. Importantly, the aim of these simulations was not to establish direct quantitative comparisons with experimental data, but rather to complement our flow cytometry-based assays by providing molecular-level insights into how the modular domain architecture of NPM1 enables condensate assembly.

NPM1 is a modular protein composed of three domains: an *N*-terminal oligomerization domain (OD, residues 1–118, red), a central intrinsically disordered region (IDR, residues 119–242, green), and a *C*-terminal DNA-binding domain (CTD, residues 243–294, blue) (Supplementary Fig. 5a, b). AlphaFold-based structure predictions revealed high-confidence folding (pLDDT > 90) for the OD and CTD, and low pLDDT scores for the IDR, indicating a flexible and disordered central region (Supplementary Fig. 5c). Notably, charge analysis revealed striking asymmetries: the OD and IDR are highly negatively charged, with net charges of −7 and −19 respectively, while the CTD is mildly positively charged (+2), consistent with its known affinity for nucleic acids (Supplementary Fig. 6a). Despite their shared negative charge, these domains differ in charge patterning. Using the Sequence Charge Decoration (SCD) parameter, which quantifies local charge clustering in disordered regions[54], we found that the IDR region has the most charge segregation (SCD = −1.473), with negative and positive charges clustered within the IDR segment (Supplementary Fig. 6a, b). Such clustering can enhance phase separation[55,56], thus motivating further exploration of how these features can potentially govern the self-assembly of NPM1.

To understand the role of these domain-specific properties in NPM1 phase separation, we simulated 100 full-length NPM1 chains using the HPS-Urry model[57] under slab geometry[58,59] to achieve coexisting dense and dilute phases. We applied rigid body motion to folded domains (OD and CTD), while disordered regions remained fully flexible (see Supporting Information for simulation details). We simulated this system at a fixed temperature of 300 K with 100 mM salt and analyzed protein densities as a function of the z-coordinate (Fig. 4b). The density profile clearly shows that the system phase separates into distinct dense and dilute phases, with NPM1 condensates forming a well-defined dense phase (Fig. 4b). To probe the molecular interactions that mediate condensate formation, we computed 2D and 1D time-averaged intermolecular contact maps across all domains (Fig. 4c, d).

Despite both the OD and IDR being negatively charged, the OD region exhibited the strongest intermolecular contacts, particularly OD–OD interactions (Fig. 4c, d and Supplementary Fig. 6c). Many residues in this region formed the highest number of intermolecular contacts (Fig. 4d). This aligns with its structural role in forming stable pentamers, as seen in the crystal structure of human NPM1 (PDB: 4N8M), and suggests that it acts as a scaffold for multivalent network formation. Interestingly, the IDR also displayed much stronger IDR–IDR interactions, despite its strong negative charge (see Supplementary Fig. 6c). This observation highlights the importance of charge patterning over net charge, as the presence of segregated blocks of charges within the IDR may facilitate transient attractions and enable phase separation. In contrast, CTD–CTD interactions were weak (Supplementary Fig. 6c), consistent with its function as a nucleic acid-binding region rather than a driver of protein–protein condensation. However, we observed favorable CTD–IDR and CTD–OD interactions, suggesting that the CTD may help stabilize the condensate by bridging

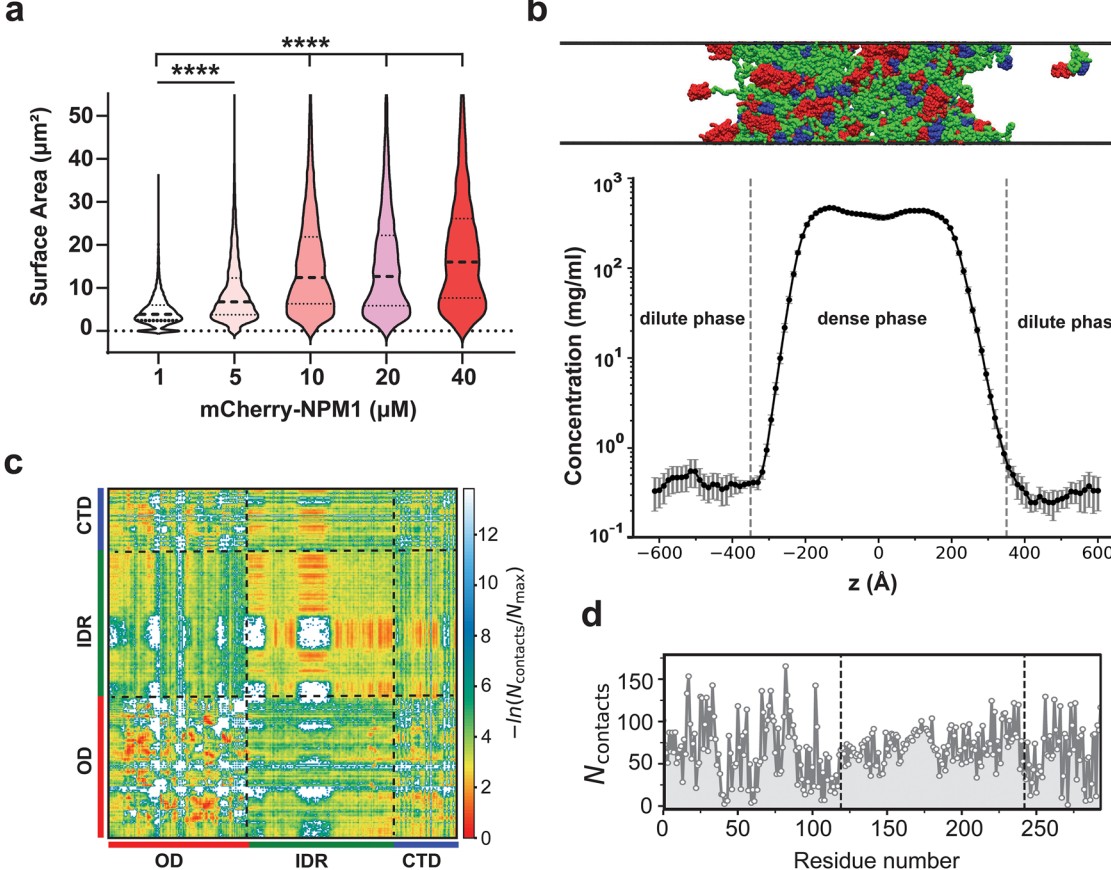

**Fig. 4 | Quantification and coarse-grained (CG) simulation analysis of NPM1 condensates. a** Quantification of condensate surface area from 10,000 individual mCherry-NPM1 condensates across protein concentrations, derived from imaging flow cytometry (IFC) images (see Supplementary Figs. 3–4). Results show a concentration-dependent increase in average condensate size. **b** Snapshot of NPM1 slab configuration (100 chains) from CG simulation in which two phases coexist. Red, green and blue colored segments represent the N-terminal OD domain, the IDR region and the C-terminal domain respectively of the full-length NPM1 protein (*Top*). Density profile of NPM1 along the z-dimension of the slab geometry. The area

between gray dashed lines denotes the dense phase and the area outside of it represents the dilute phase (*Bottom*). **c** Time-averaged intermolecular contact map of NPM1 proteins within the condensed phase. The contact number of NPM1 is normalized to the highest contact number of NPM1. Each colored bar highlights the corresponding domain of the NPM1 protein. **d** The 2D contact map shows the average number of contacts per chain as a function of residue number. Statistical analysis was performed using unpaired, two-tailed student's *t* tests. Significance is denoted as follows: $p < 0.0001$ (****). Exact p-values are provided in the Source Data file. Scale bars = 7 µm.

domains or enhancing multivalency. To further probe how electrostatics shape these interactions, we computed electrostatic surface representations of the OD and CTD (Supplementary Fig. 6d). These revealed complementary charge patches: while the OD exhibits distinct positive regions, the CTD contains negatively charged surfaces (Supplementary Fig. 6d), and the IDR features extended negative patches (Supplementary Fig. 6a). Such spatial charge complementarity provides a molecular rationale for the domain-specific contacts observed in the simulations and explains how the CTD can bridge otherwise negatively charged domains. Importantly, this electrostatic complementarity also connects to our experimental observations that increasing salt concentration disrupts condensate stability (Fig. 3f, g), as electrostatic screening weakens these multivalent charge-driven interactions.

**Colocalization study of NPM1 condensates with macromolecules**
Having established the formation and characterization of NPM1 condensates, we next aimed to assess whether flow cytometry can capture the interaction and recruitment of diverse macromolecules into these biomolecular condensates. Such condensates often function as organizational hubs, selectively incorporating proteins, nucleic acids, and small molecules according to physicochemical properties, including charge, hydrophobicity, and multivalency[23,60–63]. To assess the

partitioning behavior of different macromolecules within NPM1 condensates, we systematically examined the interaction of NPM1 condensates with antibodies, lipids, small-molecule drugs, and RNA (Fig. 5).

We first benchmarked the association of biomacromolecules with NPM1 condensates using confocal microscopy. NPM1 bearing a C-terminal His tag was probed with Alexa Fluor 488–labeled anti-HisTag (positive control) and anti-biotin (negative control) antibodies. Increasing the concentration of NPM1 led to the formation of larger condensates visible in both the mCherry and Alexa Fluor 488 channels (Fig. 5a), consistent with efficient recruitment of the anti-HisTag antibody. Flow cytometry corroborated these colocalization patterns, showing a corresponding increase in both mCherry and Alexa Fluor 488 fluorescence signals (Fig. 5b). In contrast, the anti-biotin antibody did not show co-association with NPM1 condensates, as indicated by the absence of Alexa Fluor 488 signal in both confocal imaging and flow cytometry (Fig. 5a, b). Together, these results demonstrate selective antibody partitioning with its cognate binding partner and establish flow cytometry as a complementary and quantitative method for assessing condensate–macromolecule colocalization.

Next, we examined whether flow cytometry could be used to assess the association of phospholipids with NPM1-based condensates. Recent studies have shown that phospholipids can partition into

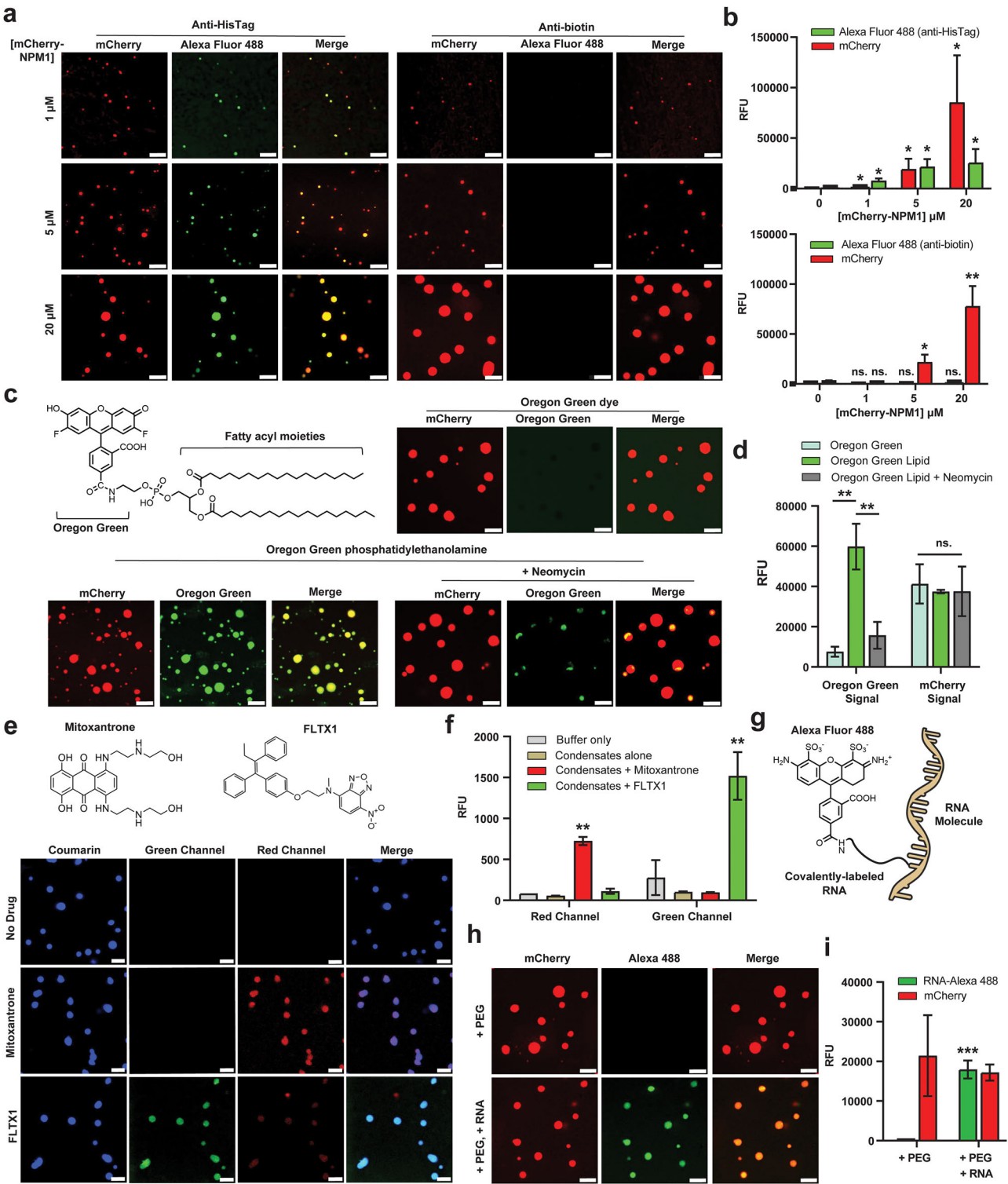

biomolecular condensates, potentially modulating their size, morphology, and function[64]. To explore this possibility, we conducted colocalization assays using two dyes: the dye Oregon Green 488 and Oregon Green–conjugated phosphatidylethanolamine, a lipid-linked dye bearing a hydrophobic tail. Consistent with previous reports, confocal imaging showed that Oregon Green 488 did not colocalize with NPM1 condensates, as only mCherry fluorescence was detected, indicating that the condensates exclude freely soluble small molecules (Fig. 5c). In contrast, Oregon Green phosphatidylethanolamine demonstrated high levels of partitioning into NPM1 condensates, indicating that lipid-modified molecules can associate with these

structures. These results mirror the previously described behavior of these two molecules with biomolecular condensates using confocal microscopy[64]. Complementary flow cytometry analysis reproduced the overall trends while enabling analysis of a much larger sample population (Fig. 5d). Alexa Fluor 488 fluorescence was low when NPM1 was incubated with Oregon Green but markedly increased with Oregon Green-tagged phosphatidylethanolamine, indicating strong partitioning of the lipid-conjugated dye into NPM1 condensates. However, when NPM1 was pre-incubated with neomycin before adding Oregon Green phosphatidylethanolamine, the Alexa 488 signal remained low, suggesting that neomycin disrupts lipid association, likely by

**Fig. 5 | Colocalization study of NPM1 condensates with macromolecules.**
**a** Confocal imaging of mCherry-NPM1 condensates incubated with Alexa Fluor 488-labeled anti-HisTag (positive control) and anti-Biotin (negative control) antibodies at different NPM1 concentrations. Anti-HisTag is selectively recruited into NPM1 condensates, while anti-biotin remains excluded. **b** Flow cytometry confirms colocalization results from **a**, showing increased Alexa Fluor 488 fluorescence for anti-HisTag but not for anti-Biotin. Statistical comparisons are made against the 0 μM protein condition. **c** Confocal imaging of 20 μM NPM1 condensates incubated with 2 μM lipid probes reveals selective partitioning of Oregon Green phosphatidylethanolamine (a lipid-modified dye) but exclusion of Oregon Green 488 (a hydrophilic dye), indicating a preference for lipid-like molecules. **d** Flow cytometry corroborates the imaging data in **c**, showing increased Alexa Fluor 488 fluorescence in the presence of Oregon Green phosphatidylethanolamine. Pre-incubation with 2 mg/mL neomycin disrupts lipid association without affecting mCherry-NPM1 condensate formation. **e** Confocal imaging of 20 μM NPM1 condensates incubated with small-molecule drugs (50 μM mitoxantrone or 100 μM FLTX1) demonstrates their strong partitioning into condensates. **f** Flow cytometry confirms drug partitioning, showing an increase in fluorescence intensity for both mitoxantrone and FLTX1 in NPM1 condensates. Statistical comparisons are between mitoxantrone versus buffer only (red channel), and FLTX1 versus buffer only (green channel). **g** Schematic representation of RNA labeling using Alexa Fluor 488. **h** Confocal imaging shows strong colocalization of RNA-Alexa 488 with NPM1 condensates, indicating RNA recruitment. **i** Flow cytometry analysis supports confocal imaging in **h**, demonstrating robust RNA partitioning into NPM1 condensates. Statistical comparisons are between condensates with RNA and condensates without RNA. Statistical analysis was performed using unpaired, two-tailed student's *t* tests. Error bars indicate standard deviation (SD). Significance is denoted as follows: $p < 0.05$ (*), $p < 0.01$ (**), $p < 0.001$ (***), $p < 0.0001$ (****) and not significant (ns.) for $p \geq 0.05$. Exact p-values are provided in the Source Data file. Scale bars = 5 μm.

interfering with electrostatic or hydrophobic interactions[65]. Meanwhile, mCherry fluorescence intensity remained high across all conditions (Fig. 5d), indicating that condensate formation itself was not affected. Collectively, these findings demonstrate that flow cytometry enables quantitative assessment of lipid–condensate interactions, providing a means to investigate how biomolecular condensates coordinate lipids in their roles as hubs of biological activity.

Likewise, prior research also highlighted that small-molecule drugs (including anti-cancer drugs) can selectively partition into biomolecular condensates, potentially influencing their local concentration and pharmacodynamics[21]. To benchmark this feature in a high-volume analysis platform, we performed colocalization assays using mitoxantrone and FLTX1, two small-molecule drugs with distinct physicochemical profiles[66,67]. Confocal imaging confirmed that both mitoxantrone and FLTX1 strongly partitioned into NPM1 condensates, as evidenced by their respective intrinsic fluorescence signals in the red and green channels, respectively (Fig. 5e). These findings were further supported by flow cytometry analysis, which showed a clear increase in drug-associated fluorescence within condensates (Fig. 5f). These results highlight the potential of flow cytometry to broadly characterize the partitioning or accumulation of small molecules within condensates in a higher-throughput manner than LC-MS/MS. This approach also avoids centrifugation steps that can disrupt condensate interfaces, enabling the analysis of intact droplets that remain as single entities in solution.

Finally, we investigated the association of RNA with NPM1 condensates, given the well-documented role of RNA in nucleating and modulating condensate formation in various cellular contexts[9]. We labeled total RNA extracted from *E. coli* with AZDye 488 (Fig. 5g) and incubated the dye-tagged RNA with mCherry-NPM1 in the presence of PEG. Confocal imaging showed strong colocalization of mCherry and AZDye 488 signals, indicating that RNA colocalizes with NPM1 condensates (Fig. 5h). Flow cytometry analysis further confirmed the localization of the two biomacromolecules (Fig. 5i), demonstrating robust RNA association with NPM1-driven condensates. We also incubated RNA with mCherry-NPM1 in the absence of PEG. Under these conditions, RNA alone could promote condensate formation (Supplementary Fig. 7a), and titration of rRNA revealed a concentration-dependent increase in mCherry fluorescence intensity (Supplementary Fig. 7b), providing a quantitative readout of RNA-induced condensate assembly. These results suggest that electrostatic interactions between positively charged regions of NPM1 and negatively charged RNA are sufficient to drive condensate formation even in the absence of crowding agents. Collectively, these colocalization studies demonstrate that NPM1 condensates selectively recruit diverse macromolecules, underscoring their role as compartmentalized microenvironments for molecular interactions. The complementary application of flow cytometry not only corroborated imaging results but also revealed subtle shifts in condensate composition, establishing

it as a robust and scalable approach for dissecting the molecular content of biomolecular condensates.

## Assessment of biomolecular condensate dynamics
Having benchmarked flow cytometry across a range of established contexts, we next aimed to uncover features of biomolecular condensates that require both finer temporal resolution and high-throughput data acquisition. To this end, we focused on the dynamic behavior of individual proteins and their potential partitioning between the dilute and dense phases. Biomolecular condensates are highly dynamic structures[68] that continuously exchange molecules with their surroundings, yet quantitatively capturing these dynamic events remains a major challenge in the field. Traditional approaches, such as confocal microscopy and FRAP, have provided valuable insights into molecular mobility within condensates[52,69,70], but they are often limited in throughput and time resolution, making it difficult to assess real-time exchange events between condensates.

To investigate exchange dynamics between condensates, we replaced mCherry with HaloTag to generate an NPM1-HaloTag fusion protein. During affinity purification, we installed three distinct chloroalkane-linked fluorescent dyes (Coumarin, R110, or TAMRA) enabling site-specific labeling of the fusion protein. While the protein was bound to the resin, fluorescent dyes were conjugated, and excess unreacted dye was subsequently washed away. This strategy ensured full removal of the unbound dye (Supplementary Fig. 8). To further confirm that fluorescence accumulation within condensates reflects specific molecular interactions rather than nonspecific partitioning of small molecules, we performed a control using small fluorescent HaloTag ligands. Pre-formed NPM1-Halo condensates were incubated with either TAMRA-chloroalkane, which covalently binds HaloTag, or a structurally similar TAMRA-alkane lacking HaloTag-binding capability. Flow cytometry analysis revealed strong TAMRA fluorescence within condensates only in the presence of TAMRA-chloroalkane, whereas TAMRA-alkane produced no detectable signal (Supplementary Fig. 9). These results demonstrate that small molecules without specific binding affinity do not spontaneously accumulate within NPM1-Halo condensates, supporting the conclusion that observed fluorescence redistribution in our system is driven by defined molecular interactions. We next performed in vitro droplet formation assays using either individual dye-labeled NPM1 variants or mixtures of differentially labeled variants to monitor condensate behavior over time. Confocal microscopy at an early point revealed that condensates exhibited distinct, non-overlapping fluorescence signals (Fig. 6a). After 30 minutes, however, most droplets exhibited complete fluorescence colocalization, indicative of molecular exchange and droplet fusion (Fig. 6a). While these results clearly demonstrate rapid molecular exchange between condensates composed of the same protein, monitoring this dynamic process in real time across large droplet

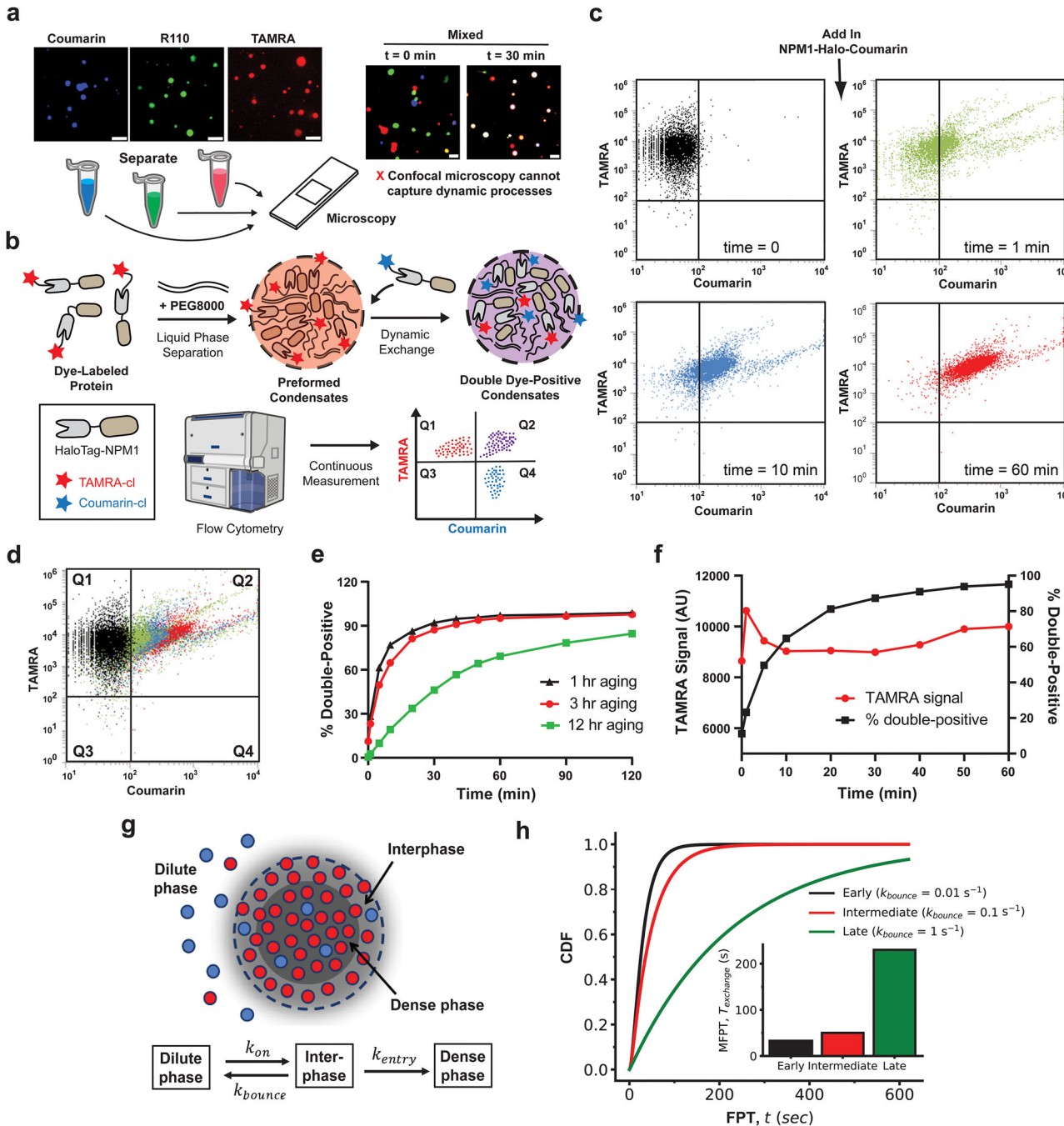

populations is challenging with confocal microscopy due to throughput limitations.

We next leveraged flow cytometry to quantitatively analyze exchange dynamics between condensates at the single-droplet level with high sensitivity and throughput. The ability of flow cytometry to rapidly measure fluorescence intensity across thousands of droplets in solution enables real-time tracking of molecular transfer between condensates. To accomplish this, we designed an assay (Fig. 6b) in which NPM1-Halo-TAMRA (red) condensates were pre-formed by equilibration with PEG. Subsequently, we introduced an NPM1-Halo-Coumarin (blue) protein solution lacking PEG and immediately monitored fluorescence over time by flow cytometry. To confirm that the observed increase in blue fluorescence within red condensates was due to active molecular exchange rather than passive mixing or dye diffusion in the bulk solution, we performed a control using free TAMRA dye. A TAMRA-only solution (without protein) was added to

PEG at the same volume ratio used in the exchange assay. After a brief 2-s mixing period, the dye rapidly dispersed and became uniformly distributed throughout the PEG solution (Supplementary Fig. 10), confirming that unbound fluorophores equilibrate freely in the solution phase. This result supports the conclusion that the observed colocalization in our primary assay arises specifically from incorporation of NPM1-Halo-Coumarin into pre-existing condensates, rather than from nonspecific mixing of fluorescent components.

To test the dynamic nature of droplet behavior, we introduced NPM1-Halo-Coumarin into pre-formed NPM1-Halo-TAMRA condensates. This resulted in rapid molecular exchange, with over 40% of events shifting to the double-positive quadrant (TAMRA+/Coumarin+) at the earliest time point measured. We note that the earliest timepoint reflects an operational dead time of approximately 10–15 s between the addition of NPM1-Halo-Coumarin, rapid mixing, tube loading, and the initiation of flow-cytometry acquisition. All kinetic assays in this

**Fig. 6 | Assessment of biomolecular condensate dynamics. a** Confocal imaging of 10 μM NPM1-Halo condensates labeled with distinct fluorescent dyes (Coumarin, R110, or TAMRA) shows that at $t = 0$, droplets remain separate, but after 30 min, they merge, indicating molecular exchange. While confocal imaging readily captures fusion endpoints, throughput limitations make it challenging to monitor large droplet populations at short time intervals. Scale bars = 5 μm. **b** Schematic of the flow cytometry-based assay for continuous real-time measurement of condensate exchange in a single-tube format, unlike conventional 96-well plate methods. Preformed NPM1-Halo-TAMRA (red) condensates were mixed with NPM1-Halo-Coumarin (blue) protein solution, and fluorescence exchange was monitored over time. **c** Flow cytometry scatter plots tracking molecular exchange. Initially, preformed NPM1-Halo-TAMRA condensates exhibit a homogeneous scatter in the TAMRA$^+$/Coumarin$^-$ quadrant. Over time, events gradually shift toward the double-positive (TAMRA$^+$/Coumarin$^+$) window, indicating progressive molecular exchange. Effective dead time from mixing to the first data acquisition is about 10–15 s; this dead time is not included in the times shown. **d** Overlay of all scatter plots from **c** shows a continuous and directional population shift toward the double-positive quadrant. **e** Quantification of exchange dynamics for 1-h, 3-h, and 12-h aged condensates. Younger condensates rapidly incorporate new proteins (>97% double-

positive events within 2 h), while older condensates show slower exchange, with 12-h aged condensates reaching only ~80%, suggesting increased rigidity or reduced mobility with aging. Each kinetic trace represents a single continuous time-course experiment ($n = 1$) for the respective aging condition. Each time point reflects 10,000 condensate events collected by flow cytometry; no error bars are shown. The 10–15 s effective dead time is not included in the times shown. **f** Monitoring of TAMRA fluorescence in 3-h aged condensates reveals a slight decrease after blue protein addition, likely due to transient redistribution or structural rearrangements. **g** Schematic of the discrete-state stochastic model for blue protein exchange into a preformed red condensate. The model considers three spatially distinct discrete states: the dilute phase (surrounding solution), the interfacial layer (condensate surface), and the dense phase (droplet interior). Proteins transition from dilute to interphase with rate $k_{on}$, return to dilute phase with rate $k_{bounce}$, or proceed into the dense phase with rate $k_{entry}$. **h** Theoretical cumulative distribution functions (CDFs) of exchange times computed under different aging conditions by varying $k_{bounce}$. Inset: Mean first-passage time (MFPT) of exchange for early ($k_{bounce} = 0.01\,s^{-1}$), intermediate ($k_{bounce} = 0.1\,s^{-1}$) and late or aged ($k_{bounce} = 1.0\,s^{-1}$) condensates. Other parameters used for the calculations are: $k_{on} = 0.05\,s^{-1}$ and $k_{entry} = 0.1\,s^{-1}$.

study use identical rapid-handling procedures, ensuring internal consistency across all experiments. Because this mixing dead time already captures a portion of the rapid NPM1 exchange process, the earliest measurement represents droplets that have begun to equilibrate. This rapid exchange raised concerns that critical transient events might be missed due to the high rate of molecular transfer. To investigate whether condensate dynamics could be artificially slowed, we explored the effect of condensate aging. Prior studies have shown that NPM1 condensates readily undergo fusion within 60 min of incubation but exhibit markedly reduced fusion rates after 180 min, likely due to aging-associated structural changes[71]. Motivated by these observations, we pre-incubated NPM1-Halo-TAMRA condensates with PEG for 1, 3, or 12 h at room temperature before the addition of NPM1-Halo-Coumarin. We then monitored 10,000 individual events over time using quadrant-based flow cytometry analysis to assess changes in exchange dynamics.

As expected prior to the addition of NPM1-Halo-Coumarin, preformed NPM1-Halo-TAMRA condensates exhibited a homogeneous scatter distribution within the TAMRA$^+$/Coumarin$^-$ quadrant (Fig. 6c). Upon addition of NPM1-Halo-Coumarin, we observed a progressive shift of events toward the double-positive TAMRA$^+$/Coumarin$^+$ quadrant, indicating active molecular exchange between the two protein pools (Fig. 6c, d). Raw scatter plots corresponding to these distributions are presented in Supplementary Fig. 11. Quantification of the percentage of double-positive events over time revealed that the exchange rate declined with increasing aging time of the pre-formed condensates (Fig. 6e), suggesting that condensate dynamics diminish as they mature. To assess how the introduction of NPM1-Halo-Coumarin impacts the structural integrity of aged NPM1-Halo-TAMRA condensates, we monitored the TAMRA fluorescence intensity in 3-h aged samples. Within 60 min of blue protein addition, the TAMRA signal exhibited a slight but consistent decrease (Fig. 6f). This reduction likely reflects the redistribution of red-labeled NPM1 due to molecular exchange. Initially, the red condensates were in equilibrium with their environment; however, the sudden influx of fresh blue protein may have perturbed this balance, resulting in transient dilution or internal rearrangements that lowered fluorescence intensity.

To further support this interpretation, we performed a control experiment in which pre-formed NPM1-Halo-TAMRA condensates were mixed with an equal concentration of soluble NPM1-Halo-TAMRA protein lacking PEG (Supplementary Fig. 12). Notably, even in this NPM1-Halo-TAMRA followed by NPM1-Halo-TAMRA condition, we observed a similar slight decrease in overall TAMRA fluorescence within 60 minutes. This suggests that the observed drop in fluorescence is not specific to dye identity or heterotypic interactions but may arise from protein influx-induced redistribution or structural remodeling of the

condensates. For the 1-h and 3-h aged samples, molecular exchange was highly efficient, with approximately 97% of condensates becoming double-positive within 2 h of mixing (Fig. 6e). In contrast, the 12-h aged sample reached only ~80% double-positive, indicating that condensate aging markedly reduces exchange efficiency. This diminished capacity may result from increased structural rigidity, reduced molecular mobility, or fibril formation that impedes protein incorporation. Supporting this, confocal imaging of 3-h aged condensates after 2 h of mixing with the blue protein showed predominantly spherical morphologies. In contrast, 12-h aged condensates exhibited irregular, broken shapes (Supplementary Fig. 13), consistent with aging-induced structural changes that impair condensate dynamics.

To further gain mechanistic insights into the dynamics of protein exchange during aging, we developed a discrete-state stochastic model that describes transitions of individual blue proteins across three spatially distinct states: the dilute phase (surrounding solution), the interfacial layer, and the dense phase (droplet interior) of a preformed red condensate, as shown schematically in Fig. 6g. This interfacial region plays a crucial role in controlling exchange kinetics, because it acts as a kinetic barrier for molecular entry[72]. In our theoretical model, blue proteins initially located in the dilute phase can enter the interfacial region with rate $k_{on}$, return or bounce back to the dilute phase with rate $k_{bounce}$, or proceed inward from interphase into the dense phase of the red condensate with rate $k_{entry}$ (see Fig. 6g). To describe the exchange dynamics of blue proteins into a preformed red condensate, we used a method of first-passage probabilities. More specifically, one can define a function $F_{dil}(t)$ as the probability density of a blue protein entering the dense phase of the red condensate for the first time at time $t$ if initially at $t = 0$, the system started in the dilute phase. A similar first-passage probability density function for the interphase can be defined as $F_{int}(t)$. The temporal evolution of these probabilities is governed by the backward master equations[73]:

$$\frac{dF_{dil}(t)}{dt} = k_{on}F_{int}(t) - k_{on}F_{dil}(t) \tag{1}$$

$$\frac{dF_{int}(t)}{dt} = k_{bounce}F_{dil}(t) + k_{entry}F_{den}(t) - (k_{bounce} + k_{entry})F_{int}(t) \tag{2}$$

where $F_{den}(t)$ is the probability density of a blue protein being found in the dense phase immediately after leaving the interphase. So, it is natural to assume that $F_{den}(t) = \delta(t)$. This physically means that if the system is in this state at $t = 0$, the process is immediately finished. As described in detail in the supporting information, these equations can be solved analytically for all ranges of parameters using Laplace transformations

$(\widetilde{F}(s) \equiv \int_0^\infty e^{-st}F(t)dt)$, producing the first-passage time (FPT) distribution $F(t)$ for entering the dense phase,

condensates accumulate proteins more slowly due to interfacial resistance and decreased internal mobility.

$$F(t) = \frac{e^{-\frac{t}{2}\left(k_{\text{on}} + k_{\text{bounce}} + k_{\text{entry}} + \sqrt{(k_{\text{on}} + k_{\text{bounce}} + k_{\text{entry}})^2 - 4k_{\text{on}}k_{\text{entry}}}\right)}\left(e^{t\sqrt{(k_{\text{on}} + k_{\text{bounce}} + k_{\text{entry}})^2 - 4k_{\text{on}}k_{\text{entry}}}} - 1\right)k_{\text{on}}k_{\text{entry}}}{\sqrt{(k_{\text{on}} + k_{\text{bounce}} + k_{\text{entry}})^2 - 4k_{\text{on}}k_{\text{entry}}}} \tag{3}$$

We also obtained the corresponding mean first-passage time (MFPT) for protein exchange, as given by

$$T_{\text{exchange}} = \frac{k_{\text{on}} + k_{\text{bounce}} + k_{\text{entry}}}{k_{\text{on}}k_{\text{entry}}} \tag{4}$$

Because the experimental setup introduces blue proteins into solution while red droplets are preformed, the system is inherently nonequilibrium as droplets can grow by recruiting newly arrived proteins. We therefore adopted kinetic rate constants from a nonequilibrium theoretical framework described by Wurtz and Lee[74], where chemical reaction-controlled exchange governs droplet formation and dynamics.

We then examined how aging affects molecular exchange by modulating $k_{\text{bounce}}$, which controls how readily proteins leave the interfacial region before entering the dense phase. This parameter is particularly sensitive to interfacial structural properties. In a recent experiment by Emmanouilidis et al.[75] that combined NMR and Raman spectroscopies with microscopy, droplet maturation or aging were investigated for FUS protein, where the structural properties upon maturation between the inside and the surface of droplets were obtained. It was found that a solid crust-like shell is observed at the surface that comprises of $\beta$-sheet contents and ultimately matured droplets were converted into fibril-like structures. This suggests that due to aging or maturation of condensates, the interface of the droplet can be much more coarsened due to this structural appearance and therefore exchange upon aging can be limited by the dynamics at the droplet interface, implying the existence of an interface resistance[72,76]. Based on these observations of interfacial solidification, we considered three conditions: "early" (low interfacial resistance, $k_{\text{bounce}} = 0.01\,\text{s}^{-1}$), "intermediate" ($k_{\text{bounce}} = 0.1\,\text{s}^{-1}$), and "late or matured" condensates (high resistance, $k_{\text{bounce}} = 1.0\,\text{s}^{-1}$). As shown in the inset of Fig. 6h, exchange becomes progressively slower with aging, with MFPT increasing from $\sim 32\,\text{s}$ (early) to $\sim 230\,\text{s}$ (late), in agreement with experimental delays in double-positive accumulation (Fig. 6e). Moreover, additional structural insights from Chatterjee et al.[77] show that interfacial regions become kinetically trapped in $\beta$-sheet–like conformations, further slowing dynamics compared to the untrapped interior. We mimicked this by decreasing $k_{\text{entry}}$ in our model for aged droplets, which also led to slower exchange times (see Supplementary Fig. 14). To further support this connection, we computed the cumulative distribution function (CDF) from the first-passage distribution:

$$\text{CDF}(t) = \int_0^t F(\tau)d\tau \tag{5}$$

which represents the probability that a protein has entered the dense phase by time $t$, and thus serves as a proxy for the fraction of blue proteins colocalized within red condensates. Under different aging conditions, the CDF curves (Fig. 6h and Supplementary Fig. 14) show a clear trend: rapid rise for early, young condensates and gradual rise for late, aged ones—qualitatively recapitulating the experimental curves in Fig. 6e. Despite the differing timescales (theoretical FPT is in seconds, and experimental mixing time is in minutes), the alignment between the theoretical and experimental dynamics strongly supports that younger condensates support faster exchange, while aged

To test whether condensate dynamics are temperature-sensitive, we performed the protein exchange assay using pre-formed, red-labeled condensates (3 h aged) and kept them on ice. Upon addition of the blue-labeled protein, minimal incorporation was observed over 90 min, with a low percentage of double-positive droplets detected by flow cytometry (Supplementary Fig. 15). However, when the same sample was returned to room temperature, a rapid increase in double-positive events was observed, eventually reaching levels comparable to our standard room temperature assay. These results suggest that lower temperatures suppress dynamic exchange, likely by reducing molecular mobility within aged condensates.

To further demonstrate the versatility of our condensate dynamics assay, we extended its application beyond protein exchange to monitor the incorporation of macromolecular cargo. Specifically, we tested whether lipid molecules could be dynamically recruited into pre-formed condensates by mixing NPM1-Halo-TAMRA condensates with Oregon Green-conjugated phosphatidylethanolamine (OG-PE). Flow cytometry analysis revealed a progressive increase in the double-positive population over 30 min (Supplementary Fig. 16), indicating successful incorporation of the fluorescent lipid into the red condensates. This observation was supported by the migration of events toward the TAMRA$^+$/OG$^+$ quadrant in the scatter plots (Supplementary Fig. 16). These results highlight that our real-time flow cytometry platform can sensitively track condensate dynamics involving diverse molecular components, offering a broadly applicable tool to uncover new principles of phase separation and inform future biological investigations.

In this study, we used NPM1 as a model scaffold protein to show that flow cytometry provides a quantitative, high-throughput method for analyzing condensate formation and macromolecular partitioning, complementing traditional imaging-based approaches. Imaging flow cytometry (IFC) further confirmed that condensates retain their structural integrity under flow, validating the reliability of flow-based analysis.

To investigate the molecular determinants of NPM1 condensate formation, we performed coarse-grained molecular dynamics simulations that revealed key domain-specific interactions driving self-association. The N-terminal oligomerization domain (OD) engaged in strong homotypic contacts, while the intrinsically disordered region (IDR)—despite its net negative charge and charge-segregated sequence—also contributed substantially to condensate stability. In contrast, the C-terminal DNA-binding domain (CTD) displayed weaker self-interactions but formed favorable contacts with both OD and IDR. These results illustrate how architectural features and sequence-level electrostatics work together to promote NPM1 self-assembly in the absence of nucleic acids or other binding partners.

We additionally developed a flow-based molecular exchange assay that revealed reduced exchange efficiency in aged condensates, likely stemming from changes in internal organization and decreased molecular mobility. To interpret these findings, we constructed a discrete-state stochastic model in which exchange proceeds through transitions between dilute, interfacial, and dense regions of a condensate. This framework captures single-molecule

exchange kinetics using FPT distributions and mean exchange times. By tuning two key parameters—the escape rate from the interface and the entry rate into the dense phase—we reproduced the experimentally observed slowing of exchange with condensate aging. The model further predicts cumulative exchange behavior through FPT analysis, closely matching experimental time courses of double-positive droplet accumulation across aging conditions. Together, these theoretical results provide a quantitative explanation for the experimental observations and highlight the importance of interfacial resistance and molecular mobility in aging-dependent exchange dynamics.

Compared to traditional microscopy-based methods, our integrated workflow yields more quantitative and mechanistic insight into condensate behavior. This platform can be readily extended to other condensate systems, including studies of how crowding agents, chaperones, or small molecules modulate condensate dynamics and stability. While flow cytometry offers robustness and scalability, it also presents challenges for future applications. Condensates are dynamic and sensitive to environmental perturbations, and shear stress or buffer composition in flow systems may subtly influence their behavior or stability. Distinguishing bona fide condensates from nonspecific aggregates also requires orthogonal validation, such as imaging or enzymatic assays. Integrating experimental measurements with molecular simulations and theoretical modeling helps mitigate these limitations, enabling a more complete and reliable understanding of condensate structure and dynamics.

In summary, this work introduces a multimodal framework for interrogating the structure, composition, and dynamics of biomolecular condensates. Flow cytometry provides a powerful quantitative complement to imaging-based approaches, molecular simulations reveal how domain-level interactions drive assembly, and a flow-based exchange assay demonstrates how aging slows molecular mobility. Together, these integrated strategies offer a unified and generalizable platform for probing condensate biophysics and their functional roles.

## Methods

### Instrumentation and reagents
DNA concentrations were quantified using a NanoDrop spectrophotometer (Thermo Fisher, CHEM-PR1-KIT). Images from agarose gels and SDS-PAGE were captured with a ChemiDoc XRS+ gel imager (Bio-Rad, 1708265). A Synergy™ H1 hybrid multi-mode reader (Agilent, BioTek Synergy H1) was employed for condensate turbidity measurement. Confocal images were obtained on a Zeiss LSM 980 Microscopy System and a Leica STELLARIS 8 confocal/FLIM/tauSTED microscope system equipped with a tunable white light laser. For most flow cytometry analyses related to condensates, an Attune™ NxT Acoustic Focusing Cytometer was used. For imaging flow cytometry (IFC) analysis, the Amnis ImageStreamX Mark II system (Luminex Corporation) was employed.

Protein expression was carried out in *Escherichia coli* BL21-Gold (DE3) competent cells (Agilent Technologies, 230132). RNA extraction was performed using *E. coli* BL21 (DE3) pLysS cells. A Monarch Total RNA Purification Kit (New England Biolabs, T2010S) was used to facilitate RNA extraction from bacterial cultures. Protein expression and purification involved the use of Isopropyl-β-D-thiogalactopyranoside (IPTG) (Chem Impex, 00194), cOmplete protease inhibitor cocktail (Millipore Sigma, 11836153001), Econo-Pac Columns (Bio-Rad, 7321010), HisPur™ Ni-NTA resin (Thermo Fisher, 88221), Amicon Ultra-15 Centrifugal Filter Unit (EMD Millipore, UFC900324), Pierce™ Slide-A-Lyzer® Dialysis Cassettes (Thermo Fisher, 66380). Chemicals used for protein and RNA labeling, as well as macromolecule compounds for condensate colocalization studies,

were obtained from commercial sources. AZDye™ 488 hydrazide (Vector Laboratories, FP-1017) was used to label bacterial RNA. Polyethylene Glycol 8000 (Sigma-Aldrich, PHR2894-1G) was used to induce condensate formation, and 1,6-Hexanediol (Sigma-Aldrich, 240117-50 G) was used to disrupt condensates. Flow cytometry assays were conducted in 96-well untreated round-bottom plates (VWR, 82050-622).

### Protein expression and purification
The plasmids encoding scaffolding proteins were acquired from Addgene: pET-45b-mCherry-NPM1 (Addgene, 194546), pET28C-mCherry-DDX4LCD (Addgene, 204408), pET-45b-mEGFP-HMGB1-WT (Addgene, 194543), pET-45b-HP1a-mCherry (Addgene, 185012). The above plasmids were a gift from Denes Hnisz from Max Planck Institute, and Samie Jaffrey from Cornell University. Codon-optimized DNA sequences of HaloTag fusion scaffolding protein (NPM1-Halo) were synthesized by GenScript and cloned into a pET-28a vector with an N-terminus His-tag. All plasmids were transformed into *E. coli* BL21-Gold (DE3) cells. The transformed cells were then inoculated into Luria-Bertani (LB) media and shaken at 37 °C until $OD_{600}$ reached 0.3–0.6. Protein expression was induced by adding 1 mM isopropyl β-D-1-thiogalactopyranoside (IPTG), followed by incubation at 37 °C for 3 h with shaking at 250 rpm. This step was followed by an extended incubation at 18 °C for 18 h. Cells were harvested by centrifugation at 4000 rpm for 30 min. The resulting cell pellets were lysed via sonication in lysis buffer (50 mM Tris-Cl, pH 7.5 at 25 °C; 1 M NaCl; 1 mM DTT) supplemented with protease inhibitors (Sigma-Aldrich, 11836153001). The lysates were centrifuged at 18,000 *g* for 20 min at 4 °C, and the clarified supernatant was applied to Ni-NTA resins (Thermo Fisher, 88221) in a gravity flow column (Bio-Rad, 7321010). The resin was allowed to bind the proteins at 4 °C overnight with rotation. The column was washed with 100 mL of wash buffer (50 mM Tris-Cl, pH 7.5 at 25 °C; 1 M NaCl; 1 mM DTT; 20 mM imidazole). Recombinant proteins were eluted with 10 mL elution buffer (50 mM Tris-Cl, pH 7.5 at 25 °C; 500 mM NaCl; 500 mM imidazole). The eluted proteins were concentrated using an ultra-15 centrifugal filter unit (EMD Millipore, UFC901024), and dialyzed into 1X storage buffer (50 mM Tris-Cl, pH 7.5 at 25 °C; 125 mM NaCl; 1 mM DTT; 10% glycerol). Protein concentrations were determined using a NanoDrop spectrophotometer (Thermo Fisher, 13-400-518), based on the calculated extinction coefficient and molecular weight of each protein. Corresponding protein sequences are provided in Supplementary Table 1.

### Protein labeling
Our lab has previously established protocols to synthesize chloroalkane ligand dyes for HaloTag binding[78]. Nonetheless, for pursuing more consistent results, we chose to purchase dyes from commercially available vendors: TAMRA-Chloroalkane (TAMRA-Cl, Promega, G8252), Coumarin-Chloroalkane (Coumarin-Cl, Promega, G8582), and R110-Chloroalkane (R110-Cl, Promega, G3221). These dyes were directly conjugated to the NPM1-Halo protein while it was bound to Ni-NTA resin, enabling efficient on-resin labeling. Briefly, clarified lysate containing NPM1-Halo was loaded onto 1 mL of Ni-NTA resin in a gravity flow column. After washing with 20 mL of wash buffer to remove non-specific proteins, the resin was incubated for 15 min at room temperature with 5 mL of 1X storage buffer containing 30 μL of 10 mM stock dye (TAMRA-Cl, Coumarin-Cl, or R110-Cl) under gentle shaking. Following labeling, the column was washed with 100 mL of wash buffer to remove unbound dye, and labeled proteins were eluted using 10 mL of elution buffer. Eluted proteins were then concentrated and dialyzed following the same protocol described in the purification section above.

## RNA isolation and labeling

*E. coli* BL21 (DE3) pLysS cells were grown at 37 °C in LB media until $OD_{600}$ reached 1.0, then centrifuged at $4000\,g$ for 30 min. All subsequent steps were performed at room temperature to prevent RNA precipitation. RNA isolation was carried out using the Total RNA Extraction kit (NEB, T2010) following the manufacturer's protocol. In brief, cells were resuspended in 1X DNA/RNA Protection Reagent with 1 mg/mL lysozyme (MP Biomedicals, 100831), followed by sonication for 30 min. The lysates were transferred to an RNase-free microfuge tube and centrifuged at $16,000\,g$ for 2 min. The supernatant containing RNA was mixed with RNA Lysis Buffer and processed according to the manufacturer's instructions. The eluted RNA was further concentrated using isopropanol and 70% ethanol, and the final RNA concentration was determined using NanoDrop. For RNA labeling, we adapted a periodate oxidation and hydrazide coupling protocol by directly oxidizing total RNA with sodium periodate followed by conjugation to AZDye 488 hydrazide[79]. Briefly, 87 µL of RNA (3.0 mg/mL) was mixed with 3.3 µL of 3 M sodium acetate (pH 5.2) and 10 µL of freshly prepared 25 mM sodium periodate. The mixture was incubated on ice for 50 min, followed by the addition of 20 µL of 3 M sodium acetate (pH 5.2) and 80 µL of nuclease-free water. RNA was precipitated by adding 400 µL of isopropanol and incubating on ice for 1 h. The RNA was pelleted by centrifugation at $16,000\,g$ for 15 min at 4 °C, then rinsed with 150 µL of ice-cold ethanol. After another centrifugation at $16,000\,g$ for 15 min, the RNA pellets were resuspended in 500 µL of reaction buffer containing 100 mM sodium acetate (pH 5.2) and 25 nmol AZDye 488 Hydrazide (Vector Laboratories, FP-1017-1). The reaction was incubated for 48 h at room temperature with gentle shaking. The labeled RNA was purified via additional isopropanol and ethanol precipitation and dissolved in 100 µL of nuclease-free water. The final RNA-AZDye488 stock concentration was determined using NanoDrop.

## In vitro droplet assays by confocal microscopy

Recombinant proteins were diluted to various stock concentrations using 1X storage buffer without PEG (Polyethylene Glycol 8000). For NaCl titration assays, the original protein stock (with 125 mM NaCl) was either diluted using 1X storage buffer with no NaCl or mixed with 1× storage buffer containing 2 M NaCl to achieve final NaCl concentration ranging from 50 mM to 500 mM before imaging. For imaging, 10 µL of fluorescent protein-fused condensate proteins were mixed with 10 µL of 20% (w/v) PEG solution in a PCR tube and allowed to incubate at room temperature for 30 min. Next, 10 µL of the protein mixture was added into a homemade flow chamber consisting of a glass slide (Corning, 2947–75 × 25) sandwiched by a coverslip (VWR, 48366-045) with a single layer of double-sided tape serving as a spacer. Images were acquired using a Zeiss LSM 980 Microscopy System, which includes an inverted Axio Observer microscope and a multiplex Airyscan module. Lasers at 405 nm, 488 nm and 567 nm were used for excitation, and appropriate PMT detectors were used for emission. As a negative control, pre-formed condensate solutions were mixed with an equal volume of 20% (w/v) 1,6-hexanediol for 10 min at room temperature before imaging. For the droplet mixing assay, a 20 µM NPM1-Halo protein solution was incubated separately with equimolar amounts of each dye (TAMRA-Cl, Coumarin-Cl, or R110-Cl) for 10 min at room temperature. The different dye-labeled NPM1-Halo proteins were then incubated with an equal volume of 20% (w/v) PEG separately for another 30 min. Subsequently, these colored NPM1-Halo condensates were combined into a single solution and placed in the flow chamber for imaging. For RNA-induced condensate formation, 20 µM mCherry-NPM1 was mixed with 50 ng/µL RNA-AZDye488 (without PEG) at room temperature for 30 min before imaging.

## Turbidity assay

To further characterize condensate formation beyond confocal microscopy, we performed turbidity measurement using mCherry-NPM1 protein. Briefly, condensate formation was assessed by mixing mCherry-NPM1 solutions with 1X storage buffer, followed by incubation with 20% (w/v) PEG for 30 min at room temperature. The resulting condensate solutions were transferred to an opaque 384-well-plate (Millipore Sigma, MZHVN0W50) in triplicate, and turbidity was measured at 600 nm using a microplate reader (Agilent, BioTek Synergy H1). The turbidity of mCherry-NPM1/PEG mixtures was analyzed across a range of mCherry-NPM1 concentrations, from 0.1 µM to 1 mM. To evaluate condensate growth over time, we prepared a mixture of 20 µM mCherry-NPM1 with 10% (w/v) PEG and monitored turbidity changes over a 30-min time course.

## Condensate imaging for colocalization assays

To assess the interaction and exchange properties of protein condensates with macromolecules in the surrounding solution, colocalization assays were performed using both confocal microscopy and flow cytometry. Colocalization was evaluated in three categories: (a) antibodies, (b) lipids, and (c) cancer drugs. For antibody-based assays, a positive control was established using a 6xHisTag monoclonal antibody conjugated to Alexa Fluor 488 (Thermo Fisher, MA1-21315-A488). A biotin monoclonal antibody labeled with Alexa Fluor 488 (Thermo Fisher, 53-9895-82) was used as a negative control. Recombinant mCherry-NPM1 protein, containing a 6xHisTag, was incubated with either anti-HisTag (1:800 dilution) or anti-biotin antibody (1:800 dilution) for 10 min at room temperature. The antibody-protein mixture was then incubated with PEG to facilitate condensate formation for 30 min, after which the samples were imaged. For lipid-based colocalization, 2 µM of Oregon Green 488 carboxylic acid (AAT Bioquest, 5-OG488) or Oregon Green phosphatidylethanolamine (Thermo Fisher, O12650) was incubated with 20 µM mCherry-NPM1 at room temperature for 10 min. The lipid-protein mixture was then combined with PEG for an additional 30-min incubation to allow condensate formation. As an additional control, 20 µM mCherry-NPM1 was pre-incubated with 2 mg/mL neomycin trisulfate salt hydrate (Millipore Sigma, N1876) for 10 min, followed by incubation with 2 µM Oregon Green phosphatidylethanolamine for 10 min. PEG was then added for another 30-min incubation prior to imaging. For cancer drug-based assays, 20 µM Coumarin-labeled NPM1-Halo protein was first incubated with PEG to form condensates. Following this, the samples were incubated with either 50 µM mitoxantrone (AmBeed, A207951) or 100 µM FLTX1 (MedChem Express, HY-119437) for 10 min at room temperature. The samples were subsequently pipetted for imaging. For confocal acquisition of drug colocalization images (Fig. 5e), samples were imaged on a Zeiss LSM 980 using 488-nm excitation with a 505–550-nm emission window for FLTX1 (green channel), and 561-nm excitation with a 590–630-nm emission window for mitoxantrone (red channel). These non-overlapping spectral windows minimized bleed-through, and single-color controls confirmed negligible cross-excitation and emission cross-talk.

## Condensate analysis by flow cytometry

To assess condensate formation and dynamics, flow cytometry was performed in parallel with confocal imaging using the same batch of protein samples, all analyzed on the same day as described above. Briefly, 35 µL of each 40 µM scaffolding protein (mCherry-NPM1, mCherry-DDX4, mCherry-HP1α, or EGFP-HMGB1) was mixed with 35 µL of either 1X storage buffer or 20% (w/v) PEG in a 96-well clear, untreated round-bottom plate (VWR, 82050-622). The mixtures were incubated at room temperature for 30 min, after which samples were analyzed on an Attune™ NxT Acoustic Focusing Cytometer. For the dissolution assay, pre-formed 20 µM condensates were treated with

10% (w/v) 1,6-hexanediol for an additional 10 min prior to flow cytometry analysis. Each condition was tested in triplicate, and the mean fluorescence intensity from over 10,000 events was used for quantification. To evaluate concentration-dependent condensate formation, varying concentrations of mCherry-NPM1 (2–80 μM) were prepared by mixing protein stock solutions with either 1X storage buffer or 20% (w/v) PEG. Samples were incubated for 30 min at room temperature before flow cytometry analysis. For the salt titration experiments, mCherry-NPM1 was diluted in 1X storage buffer supplemented with either 0 or 2 M NaCl, yielding final NaCl concentrations ranging from 100 mM to 1 M and a constant final protein concentration of 20 μM. Equal volumes of 20% (w/v) PEG were added, and condensates were allowed to form for 30 min before analysis. For colocalization assays, antibodies, lipids, and small-molecule drugs were added to mCherry-NPM1 condensates at the same concentrations used in confocal microscopy experiments. These samples were similarly incubated and analyzed by flow cytometry. For flow-cytometry acquisition of drug partitioning assays (Fig. 5f), FLTX1 fluorescence (green channel) was excited with the 488-nm laser and collected in the BL2 detector (525/50-nm bandpass), while mitoxantrone fluorescence (red channel) was excited with the 561-nm laser and collected in the YL2 detector (610/20-nm bandpass). Single-color controls were used to generate a compensation matrix, enabling correction of minor spectral spillover between the green and red channels.

To investigate condensate dynamics, a time-resolved assay was performed using NPM1-Halo. Unlike the high-throughput format, these experiments were carried out in individual tubes. Specifically, 10 μM NPM1-Halo-TAMRA was incubated with 10% (w/v) PEG for 1 to 12 h at room temperature to allow condensate formation and aging. A baseline measurement was collected at - 5 min, and soluble NPM1-Halo-Coumarin (without PEG) was added to the pre-aged NPM1-Halo-TAMRA condensates, followed by immediate vortexing for 2 s. The samples were then monitored by flow cytometry at defined time intervals to track incorporation of the coumarin-labeled protein over time.

### Imaging Flow Cytometry (IFC)

To directly visualize protein condensates and compare their morphological features across different concentrations, imaging flow cytometry (IFC) was performed using the Amnis ImageStreamX Mark II system (Luminex Corporation). mCherry-NPM1 protein samples were prepared at concentrations ranging from 1 to 40 μM and mixed with an equal volume of 20% (w/v) PEG in a 96-well plate. Following a 30-min incubation at room temperature to allow condensate formation, samples were subjected to IFC analysis. The instrument was equipped with a 60× objective and a 561 nm excitation laser for mCherry detection. Brightfield and fluorescent images of individual condensates were acquired simultaneously. A minimum of 10,000 events were collected per condition, and IDEAS® software (version 6.2, Luminex Corporation) was used for image analysis, including quantitative extraction of morphological parameters (e.g., area, circularity) and fluorescence intensity. Representative images of single mCherry-NPM1 condensates were selected across the concentration range to illustrate size and intensity changes. Surface area measurements derived from brightfield images were plotted against fluorescence intensity from the mCherry channel to assess concentration-dependent condensate growth. The combination of high-content imaging and quantitative flow analysis confirmed that condensates remain intact under flow and are amenable to both morphological and fluorescence-based characterization.

### Data processing and statistical analyses

All statistical analyses were performed using GraphPad Prism 9.5 (GraphPad Software). Bar graphs represent the mean ± standard deviation (SD) unless otherwise noted.

For conventional flow cytometry, over 10,000 events were collected per sample using the Attune™ NxT Acoustic Focusing Cytometer in volumetric mode (40 μL total acquisition volume at 200 μL/min). No gating was applied; all events were included in the analysis. The X-mean fluorescence intensity was used for quantification and plotting. Forward and side scatter (FSC/SSC) were recorded to monitor event quality, but not used for gating.

For imaging flow cytometry (IFC), >10,000 individual condensates were analyzed using the Amnis ImageStreamX Mark II system with IDEAS® software. Representative condensate images and quantitative data—such as surface area and mCherry fluorescence intensity—were extracted and displayed using scatter plots and violin plots. Condensates were identified using automated segmentation masks without manual gating or filtering. An aspect ratio versus area plot was used to identify events characteristic of condensates. Events with an aspect ratio close to 1 were interpreted as exhibiting the rounded, spherical morphology typical of biomolecular condensates.

To exclude small, spherical debris that also appeared within this region, an additional gate was defined in the top-right quadrant of the aspect ratio versus area plot, symmetrically distributed along the diagonal. This gating strategy selectively captured events with an aspect ratio of 1 and higher area values, thereby enriching for larger, condensate-like particles.

To validate this gating approach, condensates were prepared at increasing concentrations of mCherry–NPM1, and imaging flow cytometry data were acquired using an Amnis ImageStreamX Mark II. Statistical analysis of the gated population (top-right quadrant of the aspect ratio vs area plot) showed a progressive increase in mCherry fluorescence intensity with rising mCherry–NPM1 concentrations. This trend was consistent with confocal microscopy results, which demonstrated a corresponding increase in condensate size at higher protein concentrations. Notably, the fluorescence intensity increase observed within the gated population was more pronounced than that of the broader "All Events" population obtained from the fluorescence intensity versus area scatter plot.

Image-based analysis of the imaging flow cytometry data further confirmed an increase in apparent condensate size with higher mCherry–NPM1 concentrations, supporting the conclusion that the applied gating strategy effectively distinguished true condensates from background debris.

For condensate dynamic exchange assays, quadrant analysis was used to monitor the incorporation of incoming blue-labeled proteins into pre-formed red-labeled condensates. Events were plotted based on red (TAMRA) and blue (coumarin) fluorescence, and the percentage of double-positive condensates was used to quantify molecular exchange over time. Flow cytometry settings were optimized to detect biomolecular condensates and are as follows: Acquisition volume: 40 μL (200 μL/min); Stopping option: 10,000 events; Laser voltages: FSC = 100, SSC = 240, BL1 = 220, YL1 = 230, YL2 = 230, all others = 200; Threshold: SSC = 500 ($0.5 \times 10^3$).

### Coarse-Grained (CG) simulation details

We employed coarse-grained (CG) molecular dynamics simulations to investigate the phase behavior and intermolecular interactions of full-length NPM1 proteins. Each protein was modeled using a $C_\alpha$-level representation, wherein every amino acid residue is mapped to a single bead centered on its α-carbon ($C_\alpha$) position. The interactions between residues were described by the HPS-Urry model[57], which has been shown to accurately reproduce LLPS in silico[36,80]. To account for the structural organization of NPM1, we applied rigid-body motion to its folded domains—namely, the N-terminal oligomerization domain (OD; residues 14–118) and the C-terminal DNA-binding domain (CTD; residues 243–294), based on high-confidence structure predictions (pLDDT > 90). The remaining segments, including the N-terminal residues (1–13) and the central intrinsically disordered region (IDR;

residues 119–242), were treated as fully flexible. In these simulations, the OD was represented as a monomeric folded unit rather than enforcing its native pentameric state, allowing us to probe domain

After performing inverse Laplace transformation of the expression (10), we obtain the first-passage distribution function of exchange times from dilute to dense phase and it is given by

$$F(t) \equiv F_{dil}(t) = \frac{e^{-\frac{t}{2}\left(k_{on} + k_{bounce} + k_{entry} + \sqrt{(k_{on} + k_{bounce} + k_{entry})^2 - 4k_{on}k_{entry}}\right)}\left(e^{t\sqrt{(k_{on} + k_{bounce} + k_{entry})^2 - 4k_{on}k_{entry}}} - 1\right)k_{on}k_{entry}}{\sqrt{(k_{on} + k_{bounce} + k_{entry})^2 - 4k_{on}k_{entry}}} \quad (11)$$

contributions to self-association without imposing oligomeric constraints. Future work will extend these studies to explicitly include OD pentamerization.

Simulations were conducted in a rectangular slab geometry of dimensions $175 \times 175 \times 1225\,\text{Å}^3$ containing 100 NPM1 chains, with periodic boundary conditions applied in all three directions. We performed a $5\,\mu\text{s}$ long Langevin dynamics simulations at a fixed temperature of 300 K, with the friction coefficient $\gamma = m_{AA}/\tau$. Here $m_{AA}$ is the mass of each amino acid bead and $\tau$ is the damping factor set to 1000 ps. The equations of motion were integrated using a velocity-Verlet algorithm with a time step of 10 fs. All simulations were performed using the HOOMD-blue molecular dynamics engine (version 4.7) with additional features provided by the azplugins package[81]. We conducted the CG simulation at 100 mM salt concentration.

For analysis, we excluded the initial $1\,\mu\text{s}$ of the trajectory to allow for equilibration. Density profiles and intermolecular contact maps were computed from the remaining $4\,\mu\text{s}$ of production data. Error bars were calculated using block averaging by dividing the production trajectory into 4 equal blocks and computing the standard error of the mean.

### Theoretical details of first-passage calculations

Let us define $F_{dil}(t)$ as the first-passage probability density function of entering a blue protein into the dense phase of the red condensate for the first time at time $t$ if initially at $t = 0$, the system started in the dilute phase. Similarly, one can define the first-passage probability density function for the interphase as $F_{int}(t)$. Then the temporal evolution of these probability functions is governed by a set of backward master equations[73]:

$$\frac{dF_{dil}(t)}{dt} = k_{on}F_{int}(t) - k_{on}F_{dil}(t) \quad (6)$$

$$\frac{dF_{int}(t)}{dt} = k_{bounce}F_{dil}(t) + k_{entry}F_{den}(t) - (k_{bounce} + k_{entry})F_{int}(t) \quad (7)$$

In this equation, $F_{den}(t)$ is the probability density of a blue protein to be found in the dense phase immediately after leaving the interphase, and we can assume that $F_{den}(t) = \delta(t)$. This means that if the system is in this state at $t = 0$, the process is immediately accomplished.

Applying Laplace transformations $\widetilde{F}(s) \equiv \int_0^\infty e^{-st}F(t)dt$, we obtain

$$(s + k_{on})\widetilde{F_{dil}}(s) = k_{on}\widetilde{F_{int}}(s) \quad (8)$$

$$(s + k_{bounce} + k_{entry})\widetilde{F_{int}}(s) = k_{bounce}\widetilde{F_{dil}}(s) + k_{entry} \quad (9)$$

Solving Eq. (8 – 9) simultaneously, we obtain

$$\widetilde{F_{dil}}(s) = \frac{k_{on}k_{entry}}{s\left(s + k_{on} + k_{bounce} + k_{entry}\right) + k_{on}k_{entry}} \quad (10)$$

Then we can compute the cumulative distribution function (CDF) from this FPT distribution:

$$CDF(t) = \int_0^t F(\tau)d\tau \quad (12)$$

Finally, the MFPT of protein exchange from dilute to dense phase can be obtained as

$$T_{exchange} = \int_0^\infty tF_{dil}(t)dt = -\frac{\partial \widetilde{F_{dil}}(s)}{\partial s}\bigg|_{s=0} = \frac{k_{on} + k_{bounce} + k_{entry}}{k_{on}k_{entry}} \quad (13)$$

### Reporting summary

Further information on research design is available in the Nature Portfolio Reporting Summary linked to this article.

### Data availability

All flow cytometry data (FCS files) are available at Zenodo under: https://zenodo.org/records/17279422. Source data for all main-text figures and Supplementary Figs. are provided in the accompanying Source Data file. The full coarse-grained (CG) molecular dynamics trajectory files exceed public repository size limits and are therefore available under restricted access for file-size reasons; access can be obtained by contacting the corresponding author (M.M.P.) Source data are provided with this paper.

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

## Acknowledgements

This study was supported by the NIH grants 1R01AI178975-01 (M.M.P.), R35GM124893 (M.M.P.), R35GM153388 (J.M.), R01AI179080-01 (M.M.P.). We thank the W.M. Keck Center for Cellular Imaging for the usage of Zeiss LSM 980 microscopy System and Leica STELLARIS 8 confocal/FLIM/tauSTED microscope system (NIH OD030409) and thank the Flow Cytometry Core Facility for the usage of Amnis ImageStreamX Mark II system. We acknowledge the Texas A&M High Performance Research Computing (HPRC) for providing computational resources that have contributed to the results reported in the article. The content of this work is solely the responsibility of the authors and does not necessarily represent the official views of the NIH.

## Author contributions

Y.H., G.M.O., and M.M.P. conceived and designed the study. Y.H. and G.M.O. performed all flow cytometry assays, imaging flow cytometry (IFC), and data analysis. Y.H. conducted confocal imaging and turbidity measurements. Y.H. and J.Mo. carried out protein expression and purification. A.M. performed the coarse-grained (CG) molecular dynamics simulations and theoretical analyses under the supervision of J.Mi. M.M.P. provided project oversight and conceptual guidance. Y.H. and M.M.P. wrote the manuscript with input from all authors. All authors reviewed and approved the final version of the manuscript.

## Competing interests

The authors declare no competing interests.
