## [Transparent Peer Review file · Nature Communications]

A High-Throughput, Flow Cytometry Approach to Measure Phase Behavior and Exchange in Biomolecular Condensates

Corresponding Author: Dr Marcos Pires

Version 0:

Reviewer comments:

Reviewer #1

(Remarks to the Author)

This work provides a powerful framework to study dynamical behaviors of biomolecular condensates utilizing a flow cytometry. The manuscript is written clearly and the results are well-supported. The supplemental materials with the actual pictures of each step procedure would be also very valuable. However, there are several points that require further assessment before acceptance in the Nature Communications.

1. The methodology of flow cytometry for the condensates was not clear to me. For example, the buffer used in the flow cytometry was the same as the buffer for preparing condensates? Also, would the flow speed and liquid viscosity affect the morphology and dynamics of condensates?

2. The results obtained by flow cytometry need to be discussed more rigorously. In Fig. 2c, authors state that PEG treatment causes a clear shift in SSA-C versus FSC-A profiles. However, without the (overlaid) plot for the condition without PEG, it is hard to understand what kind of the shift is occurring by PEG treatment.

3. Similar line to the 2nd comment. In Fig. 3e, the authors imply that larger condensates exhibit higher light scattering (FSC-A and SSC-A) based on the scatter plot. But the distribution is not showing a simple positive correlation. Please add an explanation on the distribution pattern. Would it be helpful to look at the scatter plots of mCherry Emission versus FSC-A, for example?

4. The results of CG-MD simulation seem floating from the entire story. For example, it is difficult to find the link between the intramolecular interaction in NPM and the molecular exchange in NPM condensates. Considering the results and discussions in Fig. 6, the MD simulation would be helpful in explaining how the "aging" of the condensates mechanistically occurs at molecular level.

Reviewer #2

(Remarks to the Author)

In the manuscript "A High-Throughput, Flow Cytometry Approach to Measure Phase Behavior and Exchange in Biomolecular Condensates", He and colleagues present data from analyses of phase separation behavior using flow cytometry and compare in places to other approaches. While we already know a lot about how sequences and solution conditions affect phase behavior, it will always be relevant to have complementary techniques-in if they can be scaled to a larger number of sequences and conditions.

Overall, the paper is clearly presented, though I think the manuscript could be strengthened by putting the work in a better context (for example, what are the quantities that they are aiming to measure and what are the alternative approaches) and by quantifying the results better.

Some comments

Major:

1.
As a new method, I think it would be essential that the results are compared quantitatively to current methods including state of the art biochemical methods. How/where can I see that the results from the flow cytometry method is quantitatively the same as other quantitative measurements?

For example, statements such as “Parallel flow cytometry analysis showed an analogous increase in mCherry fluorescence intensity with increases in protein concentration (Fig. 3d), thus mirroring what was observed using confocal microscopy and consistent with a larger association of fluorescently tagged proteins per event.” should really be made more quantitative. Again, I am not sure that confocal microscopy is the state of the art for quantifying composition of condensates

2.
What is the size range of the “droplets” that are seen by the flow cytometry approach; presumably things smaller than the diffraction limit could be missed?

3.
The section “Molecular dynamics studies of NPM1” contains nice work, but I find it quite detached from the rest of the paper. The authors try to connect them in a “platform” but I don’t really see how they complement or strengthen each other. Also, there are not really any quantitative comparisons between the experiments and simulations. Maybe the authors could link the two sections better?

4.
The authors conclude “Together, these findings demonstrate that in vitro condensate formation can be reliably assessed using both traditional methods and newer approaches such as flow cytometry.”
What is the data and quantifications that support this? “Reliably assessed” is a somewhat vague statement as is “traditional methods”. Could the authors be more specific about what can be quantified, and how closely the results agree with state-of-the-art approaches. These validations are essential before the approach can be scaled to study, for example, large libraries of small molecules.

Minor:

1.
I think the authors could better explain which properties they are aiming to measure and what the alternatives might be. Also, there are a number of higher-throughput approaches available both in vitro and in vivo that might be worth mentioning.

2.
On page 5, the authors three times state that flow cytometry is superior (maybe once would be enough?), and it would be useful to be clearer in what way it is superior relative to what. The authors mostly compare to confocal microscopy, but there are many other approaches and so it would be useful to be clearer about what compromises are being made in terms of what can be measured and how accurately and rapidly.

3.
What does the sentence (p. 5) “flow cytometry can be a superior method for mimicking freedom of condensate” mean? (I am confused by the word “freedom” here)

4.
On page 8, what is meant by complexity in “consistent with increased size and complexity (Fig. 2c).”?

5.
Consider changing the word volume in “paired with high volume small molecule libraries” on p. 8 since volume here (presumably) means large number, but could also be mis-interpreted to mean a large physical volume.

6.
p. 9 “One potential concern with flow cytometry is whether shear stress or buffer conditions may compromise condensate integrity.”
Yes, or strengthen it (DOI: 10.1126/sciadv.adv7875)

7.
On p. 11, the authors write “Using the Sequence Charge Decoration (SCD) parameter, which quantifies local charge clustering in disordered regions,⁴⁸ we found that the IDR region has the most charge segregation (SCD = -1.473), with negative and positive charges clustered within the IDR segment (Fig. S4a-b).”
Does it make sense to compare the values of SCD of the IDR with those of the OD and CTD?

8.
In the simulations of the full length NPM1, is the OD kept “monomeric” and if so why?

9.
In the sentence (p. 15) "These findings support the idea that biomolecular condensates can selectively incorporate lipid-like molecules, potentially modulating their biochemical properties." it is unclear whether "their" refers to the condensates, lipids or both.

10.
On page 16, the authors write "they were noticeably smaller and less spherical compared to those formed with PEG (Fig. 5h)" and "the resulting structures are substantially smaller, highlighting the role of crowding agents like PEG in enhancing condensate growth and detectability in vitro." Can these observations be quantified?

11.
Page 21 "fibrillation" -> fibril formation

12.
In the equations on p. 22 (bottom) and p. 23 (top), is there a "d" missing on the left-hand side?

13.
Figure 1 "Macromoleclues" -> "Macromolecules"

Reviewer #3

(Remarks to the Author)

The authors present a flow cytometry-based framework for high-throughput, quantitative analysis of biomolecular condensate formation, composition, and molecular exchange. In general, they confirm that assay readouts can be obtained by FCM with statistical robustness due to high throughput. I like the approach of repeating high-throughput measurements at high frequency to increase temporal sampling. However, the claims and wording need to be much more precise.

The interesting demonstration is limited in Figure 6. Other parts, while methodologically sound, are overstated or conflate concepts such as resolution, temporal resolution, and signal-to-noise ratio. For example, the manuscript repeatedly uses "real-time," "the first assay," "temporal resolution," and "superior signal-to-noise ratio" without adequate evidence or careful qualification. Microscopy and microfluidic approaches have been used to measure condensate exchange in real time (though at lower throughput), so the "first assay" claim is overreaching. The actual advantage here is high-throughput, repeated sampling across many droplets, not tracking the same objects over time.

I cannot recommend this manuscript for publication. The method has merit for high-throughput screening of condensate properties, but claims must be moderated, missing controls and gating information added, and the text revised for conceptual accuracy. Also, since the demonstration does not prove its significance, it is appreciated if further demonstrations which cannot be done microscopically.

Specific Comments:

"The first assay" in the Abstract is an overclaim. Real-time exchange measurements have been reported using microscopy and microfluidic devices, albeit at lower throughput. Reflect the specific novelty of your FCM-based approach.
Fig. 2c and Supp. Fig. S1: The claimed SSC-A vs FSC-A shift for PEG treatment is not clearly visible. Proper controls must be shown.

Fig. 2d: The claim that such resolution is difficult to achieve by microscopy is misleading; microscopy offers fundamentally higher spatial resolution, and methods such as 3D quantitative phase imaging can provide more detailed scattering and morphology. Do not conflate spatial resolution (microscopy) with statistical resolution (FCM). FCM excels in sample size and scalability; microscopy excels in spatial detail and single-object tracking. In addition, FSC vs fluorescence plots would be more informative than the violin plot.

Fig. 4a,b: The rationale for using image FCM data in the main text is unclear; they could be moved to Supplementary.

Fig. 6 caption: The statement about confocal lacking temporal resolution is misleading. The process observed is not inherently too fast; rather, throughput limits prevent measurement of large droplet populations at 1 min intervals. This should be stated accurately.

The "superior signal-to-noise" claim (p. 5) is unsupported and likely incorrect without quantitative comparison.

Is "Thousands to millions of droplets within minutes" true for all the instruments used in this study, including imaging FCM; clarify. FCM analysis lacks presentation of gating strategies; these should be included. For transparency, raw FCS data should be deposited in an open repository.

Version 1:

Reviewer comments:

Reviewer #1

(Remarks to the Author)

The authors responded well to the 3 reviewers' comments. I agree that the paper is accepted.

Reviewer #2

(Remarks to the Author)

In the revised version of the manuscript "A High-Throughput, Flow Cytometry Approach to Measure Phase Behavior and Exchange in Biomolecular Condensates", He and colleagues address many of my comments. Further, their answers should — together with my original comments — be sufficient for the reader to form their own opinion about the robustness of the work.

Reviewer #3

(Remarks to the Author)

The authors have substantively addressed prior concerns by tempering novelty claims, clarifying the complementary roles of microscopy and FCM, improving figure clarity (Fig. 2 and Fig. 6), and enhancing transparency (gating; data deposition). The study will deliver a robust, scalable platform for population-level condensate kinetics. Accept after the following minor revision:

1. How much "real-time" it is. can authors report the effective dead time from mixing to the first FCM acquisition (e.g., mixing, transfer, start of read). In Fig. 6c–e (pp. 47–48), the earliest point already shows >40% double-positive events, implying that sub-minute exchange may be under-sampled.
2. For multicolor/drug assays in Fig. 5e–f, specify the filters/compensation matrix (briefly is enough) so readers can assess potential color leaks and reproduce the settings.
3. In Fig. 6e, state replicate number (n) and error-bar definition (mean \pm SD or SEM) in the caption or Methods?

Reviewer 1

Comment 1: The methodology of flow cytometry for the condensates was not clear to me. For example, the buffer used in the flow cytometry was the same as the buffer for preparing condensates? Also, would the flow speed and liquid viscosity affect the morphology and dynamics of condensates?

Response: We thank the reviewer for raising these important methodological points. The condensates were prepared in storage buffer (50 mM Tris-Cl, pH 7.5 at 25°C; 1 M NaCl; 1 mM DTT) supplemented with 10 % PEG8000. During flow cytometry analysis, the sample started in this same buffer, while the instrument sheath fluid (a standard proprietary solution provided by the manufacturer but we have found it to be equivalent to PBS) is designed to serve for hydrodynamic focusing and does not mix with the sample stream. Therefore, the buffer environment of the condensates is likely to remain unchanged throughout the analysis. Regarding the potential effects of flow speed and liquid viscosity, we used standard flow rates to facilitate widespread adoption of this method for condensate studies. The hydrodynamic focusing employed by the flow cytometer ensures laminar flow, and the transit time through the laser interrogation point is very brief. Under these conditions, shear forces are minimal and we projected that they would not disrupt condensate morphology (imaging flow cytometry confirmed this). While we acknowledge that extreme flow rates could potentially compromise condensate integrity, our imaging flow cytometry (IFC) analysis demonstrates that consistent condensate morphology, scatter, and fluorescence signals are maintained under standard operating conditions (**Fig.4**).

Comment 2: The results obtained by flow cytometry need to be discussed more rigorously. In Fig. 2c, authors state that PEG treatment causes a clear shift in SSA-C versus FSC-A profiles. However, without the (overlaid) plot for the condition without PEG, it is hard to understand what kind of the shift is occurring by PEG treatment.

Response: We thank the reviewer for highlighting the need for clearer presentation of our flow cytometry data. To address this concern, we have revised both our figures and the main text discussion. Specifically, we now present side scatter (SSC) versus forward scatter (FSC) plots for mCherry-NPM1 under both no-PEG and +PEG conditions (**Fig. 2c**), which reveal a dramatic increase in event counts (9,567 vs. 765) and a clear redistribution of particle populations upon PEG treatment. Because light scattering depends on size, refractive index, and internal structure, FSC and SSC are best regarded as optical proxies rather than absolute measures of dimension, and we have explicitly included this clarification in the revised manuscript. We further complemented this analysis with histogram plots of mCherry fluorescence intensity (**Fig. 2d**), which show a distinct rightward shift under PEG conditions. Additional scatter and histogram data for other proteins are now included in **Fig. S1**. Together, these revisions provide a more rigorous and transparent presentation of the cytometry data, highlighting its statistical power to capture PEG-induced condensate formation.

Comment 3: Similar line to the 2nd comment. In Fig. 3e, the authors imply that larger condensates exhibit higher light scattering (FSC-A and SSC-A) based on the scatter plot. But the distribution is not showing a simple positive correlation. Please add an explanation on the distribution pattern. Would it be helpful to look at the scatter plots of mCherry Emission versus FSC-A, for example?

Response: We agree with the reviewer that the relationship between condensate size and light scattering (FSC-A and SSC-A) is not strictly linear nor do we claim this to be the case. As shown in **Fig. 3e**, the distribution pattern indicates that while larger condensates generally exhibit higher scattering signals, the data also reflect heterogeneity in condensate morphology, internal structure, and refractive index. This non-linear relationship is consistent with Mie scattering theory, which predicts that scattering intensity depends on both particle size and refractive index contrast in a complex, non-monotonic manner rather than scaling linearly with size (*Cytometry A* 97, 569, 2020; *Ultrasound Med. Biol.* 40, 138, 2014). The spread observed in the distribution likely reflects contributions from condensate shape irregularities and packing density differences, which can modulate scattering signals even at similar sizes. In response to the reviewer's suggestion, we further analyzed scatter plots of mCherry emission versus FSC-A at different protein concentrations (Figure R1). While the shifts in these plots are less pronounced than those in **Fig. 3e**, we still observed a modest trend toward higher FSC-A and fluorescence signals at elevated concentrations. We attribute the weaker separation to the fact that fluorescence intensity reports on protein content within condensates, whereas scattering signals are additionally influenced by shape irregularities, packing density, and local refractive index variations. These factors collectively blur a strict one-to-one correlation but remain consistent with the heterogeneous nature of biomolecular condensates. We have incorporated this explanation and the new analyses into the revised manuscript.

Figure R1. Scatter plots of mCherry fluorescence intensity versus forward scatter area (FSC-A) for mCherry-NPM1 at increasing protein concentrations (1, 5, 10, 20, and 40 μ M). A modest trend toward higher FSC-A and fluorescence intensity is observed at elevated concentrations, consistent with increased condensate size and protein content per particle. The distributions do not follow a strict linear shift, reflecting heterogeneity in condensate morphology, internal packing density, and refractive index, which influence scattering signals in addition to size.

Comment 4: The results of CG-MD simulation seem floating from the entire story. For example, it is difficult to find the link between the intramolecular interaction in NPM and the molecular exchange in NPM condensates. Considering the results and discussions in Fig. 6, the MD simulation would be helpful in explaining how the "aging" of the condensates mechanistically occurs at molecular level.

Response: We appreciate the reviewer's comment and understand the concern. We would like to clarify that in our study, the coarse-grained molecular dynamics (CG-MD) simulations and the theoretical model address complementary but distinct aspects of condensate behavior. The CG-MD simulations were designed to reveal how NPM1 alone can organize into phase-separated assemblies, with a specific focus on the contribution of its structural domains (OD, IDR, CTD) to self-association and condensate stability in the absence of nucleic acids or other binding partners. In contrast, the stochastic theoretical framework in **Fig. 6** was developed to rationalize the experimentally observed exchange kinetics and their modulation by aging. In particular, the theoretical model allowed us to explicitly test how interfacial resistance and altered entry rates can slow down exchange, thereby providing a mechanistic rationale for the observed experimental delays.

Thus, while both approaches provide mechanistic insight into condensate behavior, they were not intended to be directly linked at the level of molecular exchange. Instead, they collectively broaden the scope of our study: the simulations highlight the architectural and sequence-level determinants of NPM1 self-assembly, while the theory accounts for the dynamic exchange properties observed experimentally. Together, these complementary perspectives contribute to a more complete understanding of condensate formation and dynamics.

With respect to the reviewer's suggestion, we agree that a detailed molecular-level description of how condensate aging occurs mechanistically would be highly valuable. However, probing the liquid-to-solid transitions that underlie condensate maturation (such as fibril-like structural conversions observed experimentally; *Nat. Chem. Biol.*, 20, 1044, 2024) requires simulation timescales that are not feasible with our current CG-MD approach. As highlighted in a recent review (*Curr. Opin. Chem. Biol.*, 75, 102333, 2023), capturing such slow, nonequilibrium transitions by molecular simulation remains a fundamental challenge. For this reason, we instead employed a theoretical framework to model aging at the kinetic level. Molecular-level mechanisms of condensate aging are the subject of ongoing investigations in our laboratory and will be pursued in future work.

Reviewer 2

Major Comment 1: As a new method, I think it would be essential that the results are compared quantitatively to current methods including state of the art biochemical methods. How/where can I see that the results from the flow cytometry method is quantitatively the same as other quantitative measurements? For example, statements such as “Parallel flow cytometry analysis showed an analogous increase in mCherry fluorescence intensity with increases in protein concentration (Fig. 3d), thus mirroring what was observed using confocal microscopy and consistent with a larger association of fluorescently tagged proteins per event.” should really be made more quantitative. Again, I am not sure that confocal microscopy is the state of the art for quantifying composition of condensates

Response: We thank the reviewer for this important suggestion. We agree that providing quantitative comparisons between our flow cytometry approach and established methods is critical for validating our methodology. We quantified confocal images of mCherry-NPM1 condensates (n = 3 independent fields per concentration; >10 droplets per field) in ImageJ by segmenting spherical droplets and measuring mean fluorescence intensity (MFI) per droplet. To avoid pseudoreplication, we computed the image-level observations for statistical comparisons. MFI increased systematically with protein concentration, and the confocal-derived image medians correlated strongly with flow-cytometry median fluorescence measured on the same samples (Pearson's $r = 0.9653$). These results are shown in the revised manuscript, along with more representative confocal images (new **Fig. S2**). Additionally, we note that the same concentration-dependent trend was independently observed in turbidity measurements of mCherry-NPM1 condensates (**Fig. 3a**), further confirming that all three orthogonal methods (confocal imaging, flow cytometry, and turbidity) consistently report the increase in condensate abundance and fluorescence intensity with protein concentration. Beyond performing these new quantitative comparisons, we have carefully revised the main text to address the reviewer's concerns about overstatement. The revised text now highlights that confocal microscopy remains valuable for assessing morphology and spatial distribution, while flow cytometry could potentially rapid, population-level quantification across a larger throughput of individual events as a complementary tool.

New Figure S2. Quantitative comparison of confocal imaging and flow cytometry analysis of mCherry-NPM1 condensates. **a)** Representative confocal fluorescence images of mCherry-NPM1 condensates at 1, 5, 10, 20, and 40 μM protein concentration. For each concentration, three independent fields of view are shown. Scale bar = 5 μm . **b)** Quantification of mean fluorescence intensity (MFI) per condensate droplet from confocal images using ImageJ segmentation. Data was pooled from three independent fields of view per concentration ($n > 30$ condensates in total; >10 condensates per field). **c)** Correlation between confocal-derived image medians (panel **b**) and median fluorescence measured by flow cytometry on the same samples (**Fig. 3d**). Pearson's $r = 0.9653$, $p = 0.0077$ (**), indicating a strong and statistically significant positive correlation between the two methods.

Major Comment 2: What is the size range of the “droplets” that are seen by the flow cytometry approach; presumably things smaller than the diffraction limit could be missed?

Response: We thank the reviewer for raising this important question. Indeed, conventional flow cytometers have a practical lower detection limit of approximately 300–500 nm for polystyrene beads, depending on optical settings (*Cytometry B Clin. Cytom.*, 90, 326, 2016). Because biomolecular condensates typically have lower refractive index contrast than polystyrene, the effective threshold for reliable detection is likely somewhat higher. However, in our assays, the condensates visualized by confocal microscopy are predominantly several micrometers in diameter (**Fig. 2 and 3**), which is well above this detection range. Thus, the droplets analyzed by flow cytometry correspond to the main condensate population observed by microscopy methods.

Furthermore, we note that FSC and SSC signals are not direct readouts of particle size. Scatter intensity is reported in arbitrary units and depends not only on diameter but also on refractive index, morphology, and instrument optics. Light scatter sensitivity is typically described only in terms of the smallest detectable polystyrene bead, which specifies the detection threshold but provides no absolute size calibration (*Cytometry* 99, 671, 2020). For this reason, we rely on orthogonal imaging to confirm condensate sizes rather than inferring absolute diameters from FSC/SSC alone. We have revised the main text to include this discussion.

Major Comment 3: The section “Molecular dynamics studies of NPM1” contains nice work, but I find it quite detached from the rest of the paper. The authors try to connect them in a “platform” but I don’t really see how they complement or strengthen each other. Also, there are not really any quantitative comparisons between the experiments and simulations. Maybe the authors could link the two sections better?

Response: We thank the reviewer for this thoughtful comment. We want to highlight here that our molecular dynamics (MD) section is not intended to provide direct quantitative comparisons with the experiments. Rather, our aim in including the MD simulations was to uncover how the modular domain organization of NPM1 contributes to its self-association and assembly into condensates. Specifically, the MD simulations provide structural and mechanistic insights into the relative roles of the N-terminal oligomerization domain, intrinsically disordered regions, and DNA recognition motifs in driving multivalent interactions. These molecular-level insights are not directly accessible from the flow cytometry-based experimental measurements, but they establish a complementary perspective that explains why NPM1 forms condensates and how its distinct domains contribute to phase separation. To make this intent clearer, we have revised the MD section to explicitly highlight its complementary role and to avoid the impression that a direct quantitative experiment–simulation comparison was intended (see page 13).

To further strengthen the connection between simulations and experiments, we additionally analyzed the electrostatic surface potentials of the OD and CTD domains (new **Fig. S6d**). These surfaces revealed complementary charge distributions: the OD and CTD harboring positive and negative patches, respectively, while the IDR contains extended negative regions. Such electrostatic complementarity provides a molecular rationale for the domain-specific contacts observed in our simulations and directly links to our experimental findings that condensates are disrupted by increased salt concentration (**Fig. 3f–g**). By linking domain-resolved electrostatics with salt-dependent condensate stability, the revised text now integrates the experimental and simulation sections into a unified framework (see page 15).

Major Comment 4: The authors conclude “Together, these findings demonstrate that in vitro condensate formation can be reliably assessed using both traditional methods and newer approaches such as flow cytometry.” What is the data and quantifications that support this? “Reliably assessed” is a somewhat vague statement as is “traditional methods”. Could the authors be more specific about what can be quantified, and how closely the results agree with state-of-the-art approaches. These validations are essential before the approach can be scaled to study, for example, large libraries of small molecules.

Response: We thank the reviewer for pointing out this vague phrasing. We agree that the statements “reliably assessed” and “traditional methods” lacked specificity. To address this, we have revised the sentence to directly reflect our data. In our study, turbidity measurements demonstrate increased OD₆₀₀ with higher protein concentrations, while confocal microscopy shows a corresponding increase in condensate size (**Fig. 3a-d**). To further validate the flow cytometry approach, we quantified mean fluorescence intensity (MFI) from >30 condensates imaged by confocal microscopy (using ImageJ) and compared these values with MFI obtained by flow cytometry. This analysis revealed a strong correlation between the two methods (Pearson's $r = 0.9653$, new **Fig. S2**). Together, these results provide quantitative evidence that flow cytometry yields measurements consistent with confocal microscopy, supporting its use for scalable studies. We have revised the main text to clarify these quantifications and added **Fig. S2** to present the correlation analysis.

Minor Comment 1: I think the authors could better explain which properties they are aiming to measure and what the alternatives might be. Also, there are a number of higher-throughput approaches available both in vitro and in vivo that might be worth mentioning.

Response: We thank the reviewer for this helpful suggestion. Our study focuses on measuring condensate size, morphology, and fluorescence intensity, as these properties reflect both the physical state and molecular composition of condensates. To clarify this point, we have revised the Introduction to better define the properties being measured in this work. Additionally, while our original manuscript discussed several in vitro approaches (e.g., FRAP, turbidity, and sedimentation assays), we now expand the discussion to include emerging in vivo tools. Specifically, we highlight live-cell imaging and optogenetic systems that enable controlled condensate assembly and dynamic tracking, while also noting their current limitations in throughput and quantitative precision. These revisions provide clearer context for our choice of flow cytometry and situate our approach among available experimental strategies.

Minor Comment 2: On page 5, the authors three times state that flow cytometry is superior (maybe once would be enough?), and it would be useful to be clearer in what way it is superior relative to what. The authors mostly compare to confocal microscopy, but there are many other

approaches and so it would be useful to be clearer about what compromises are being made in terms of what can be measured and how accurately and rapidly.

Response: We thank the reviewer for this thoughtful feedback. We agree that our original phrasing overstated the point by repeatedly claiming that flow cytometry is "superior." We have revised the text to clarify the specific advantages that flow cytometry provides. Specifically, we now emphasize that flow cytometry offers higher throughput, better statistical power, and reduced photobleaching compared to confocal microscopy. At the same time, we acknowledge that confocal imaging retains advantages in spatial resolution and direct visualization of condensate morphology, while other assays (e.g., FRAP) provide complementary insights into dynamics and composition. These revisions make the trade-offs clearer and better situate flow cytometry among existing methods.

Minor Comment 3: What does the sentence (p. 5) "flow cytometry can be a superior method for mimicking freedom of condensate" mean? (I am confused by the word "freedom" here)

Response: We thank the reviewer for pointing out the confusing phrasing. The original phrase "freedom of condensate" was intended to convey that flow cytometry allows condensates to remain freely suspended in solution rather than being constrained on a surface or in a small imaging volume. We have revised the text to clarify this point and now explicitly state that flow cytometry enables high-throughput and quantitative analysis while maintaining condensates in their native, unconfined state.

Minor Comment 4: On page 8, what is meant by complexity in "consistent with increased size and complexity (Fig. 2c)."?

Response: We thank the reviewer for highlighting the ambiguous use of "complexity." Here, we refer to the internal granularity or structural heterogeneity of condensates, which is captured by side scatter (SSC) in flow cytometry. We have revised the text to replace "complexity" with "internal granularity" and clarified how SSC and FSC relate to particle properties, while maintaining the caution that these are optical proxies rather than direct size measurements.

Minor Comment 5: Consider changing the word volume in "paired with high volume small molecule libraries" on p. 8 since volume here (presumably) means large number, but could also be mis-interpreted to mean a large physical volume.

Response: We thank the reviewer for this suggestion. To avoid potential confusion, we have replaced "high volume" with "large-scale" to clearly indicate a large number of small molecules rather than physical volume.

Minor Comment 6: p. 9 “One potential concern with flow cytometry is whether shear stress or buffer conditions may compromise condensate integrity.” Yes, or strengthen it (DOI: 10.1126/sciadv.adv7875)

Response: We thank the reviewer for this suggestion. We have added a sentence citing Chauhan et al., 2025, to highlight that active transport and dynamic cellular processes can reinforce condensate stability, indicating that physical stresses such as shear or buffer conditions do not necessarily compromise integrity. While this may be the case in a situation of a longer time scale, given the brevity of the transit from the sample solution to the analysis we do not believe this will lead to general alteration of the output observed.

Minor Comment 7: On p. 11, the authors write “Using the Sequence Charge Decoration (SCD) parameter, which quantifies local charge clustering in disordered regions,⁴⁸ we found that the IDR region has the most charge segregation (SCD = -1.473), with negative and positive charges clustered within the IDR segment (Fig. S4a-b).” Does it make sense to compare the values of SCD of the IDR with those of the OD and CTD?

Response: We understand the reviewer’s concern. We agree that the SCD parameter is most meaningful when applied to intrinsically disordered regions, as it was originally developed to capture charge clustering in flexible segments rather than well-folded domains. Our goal in presenting the SCD values for the OD, IDR, and CTD was to provide a comparative view of charge distribution across the full NPM1 sequence. While the OD and CTD, being largely folded domains, are not ideally suited for SCD-based analysis, their values were included as qualitative indicators to highlight relative differences in charge patterning among the three domains. In this context, the comparison serves to emphasize that NPM1 exhibits distinct sequence-level asymmetries, with the IDR displaying the strongest charge clustering which is likely to influence condensate formation.

Minor Comment 8: In the simulations of the full length NPM1, is the OD kept “monomeric” and if so why?

Response: We thank the reviewer for raising this important point. In our coarse-grained simulations, the OD was modeled as a rigid folded domain but treated as monomeric rather than enforcing its pentameric state. The OD was kept monomeric to evaluate how the different domains of NPM1 contribute to self-association in the absence of predefined oligomeric constraints. By not imposing pentamerization, the simulations could probe how the OD interacts with other domains and contributes to condensate formation in a more unbiased manner. Notably, even under this setup, the OD exhibited strong intermolecular contacts consistent with its known propensity to oligomerize, underscoring its dominant role as a driver of NPM1 self-association. We also note that exploring the effect of OD pentamerization on condensate behavior is an important next step, and ongoing work in our laboratory is directed toward such pentamer-based simulations. For clarity, this modeling choice has now been explicitly noted in the Supporting Information (CG Simulation Details section).

Minor Comment 9: In the sentence (p. 15) “These findings support the idea that biomolecular condensates can selectively incorporate lipid-like molecules, potentially modulating their biochemical properties.” it is unclear whether “their” refers to the condensates, lipids or both.

Response: We thank the reviewer for this suggestion and have clarified the sentence to specify that “their” refers to the condensates.

Minor Comment 10: On page 16, the authors write “they were noticeably smaller and less spherical compared to those formed with PEG (Fig. 5h)” and “the resulting structures are substantially smaller, highlighting the role of crowding agents like PEG in enhancing condensate growth and detectability in vitro.” Can these observations be quantified?

Response: We thank the reviewer for this suggestion. In response, we have now included RNA titration experiments where increasing rRNA concentrations were added to mCherry-NPM1 in the absence of PEG, and the resulting condensate formation was quantified using flow cytometry (new **Fig. S7b**). These data show a clear, concentration-dependent increase in median mCherry fluorescence intensity, providing quantitative evidence that RNA promotes condensate formation. We have revised the text to highlight this trend while avoiding direct comparisons of condensate size and shape between PEG- and RNA-driven conditions. This allows us to present a quantitative readout of RNA-induced condensate assembly without overinterpreting differences in morphology, consistent with the original intent of using these experiments as a proof of principle for flow cytometry–based detection of condensate composition.

Minor Comment 11: Page 21 “fibrillation” -> fibril formation

Response: We thank the reviewer for the suggestion and have replaced “fibrillation” with “fibril formation” in the manuscript.

Minor Comment 12: In the equations on p. 22 (bottom) and p. 23 (top), is there a “d” missing on the left-hand side?

Response: We thank the reviewer for identifying this mistake. Yes, there is a missing “d” on the left-hand side of both the equations which we corrected now in both the revised manuscript and the supporting information.

Minor Comment 13: Figure 1 “Macromoleclues” -> “Macromolecules”

Response: We thank the reviewer for noticing this typographical error and have corrected “Macromoleclues” to “Macromolecules” in **Fig. 1**.

Reviewer 3

Comment 1: “The first assay” in the Abstract is an overclaim. Real-time exchange measurements have been reported using microscopy and microfluidic devices, albeit at lower throughput. Reflect the specific novelty of your FCM-based approach. Fig. 2c and Supp. Fig. S1: The claimed SSC-A vs FSC-A shift for PEG treatment is not clearly visible. Proper controls must be shown.

Response: We thank the reviewer for this comment. We have removed the phrase “first assay” from the Abstract to avoid overclaiming and now describe our method as a high-throughput flow cytometry approach to measure real-time exchange of biomolecules in condensates. Regarding the SSC-A vs FSC-A analysis, we have revised both our figures and the main text to improve clarity. Specifically, **Fig. 2c** now displays side-by-side scatter plots for mCherry-NPM1 with and without PEG treatment, showing a marked increase in event counts and a clear redistribution of populations under PEG conditions. In addition, we included complementary histogram plots of fluorescence intensity (**Fig. 2d**) and updated Supplementary **Fig. S1** with overlaid scatter and fluorescence distribution plots for the other proteins. Together, these revisions provide the appropriate controls and better highlight the PEG-induced shifts in the flow cytometry data.

Comment 2: Fig. 2d: The claim that such resolution is difficult to achieve by microscopy is misleading; microscopy offers fundamentally higher spatial resolution, and methods such as 3D quantitative phase imaging can provide more detailed scattering and morphology. Do not conflate spatial resolution (microscopy) with statistical resolution (FCM). FCM excels in sample size and scalability; microscopy excels in spatial detail and single-object tracking. In addition, FSC vs fluorescence plots would be more informative than the violin plot.

Response: We thank the reviewer for this comment. We have revised the text to clarify that flow cytometry provides high-throughput quantitative measurements and statistical resolution across large numbers of condensates, whereas microscopy offers higher spatial resolution and single-object tracking. In addition, we have supplemented the analysis with FSC vs fluorescence scatter plots to better visualize population-level trends, as now shown in new **Fig. S1**.

Comment 3: Fig. 4a,b: The rationale for using image FCM data in the main text is unclear; they could be moved to Supplementary.

Response: We thank the reviewer for this suggestion. We have moved the imaging flow cytometry (IFC) panels (originally **Fig. 4a-b**) to Supplementary Information (new **Fig. S3**), while retaining the quantitative analysis of condensate size (**Fig. 4c**) in the main text. The main text now refers to the representative IFC images and scatter plots in the Supplementary Information.

Comment 4: Fig. 6 caption: The statement about confocal lacking temporal resolution is misleading. The process observed is not inherently too fast; rather, throughput limits prevent measurement of large droplet populations at 1 min intervals. This should be stated accurately.

Response: We thank the reviewer for this comment. We have revised the caption of **Fig. 6** and the corresponding main text to clarify that confocal microscopy readily captures fusion events, but its throughput limits make it less suited for monitoring real-time exchange across large droplet populations.

Comment 5: The “superior signal-to-noise” claim (p. 5) is unsupported and likely incorrect without quantitative comparison.

Response: We thank the reviewer for this comment. We have removed the signal-to-noise ratio statement and revised the text (p. 5) to highlight other demonstrated advantages of flow cytometry, including reduced photobleaching, minimized user bias, reproducibility, and high throughput.

Comment 6: Is “Thousands to millions of droplets within minutes” true for all the instruments used in this study, including imaging FCM; clarify. FCM analysis lacks presentation of gating strategies; these should be included. For transparency, raw FCS data should be deposited in an open repository.

Response: We thank the reviewer for this important point. We have clarified in the main text that the statement “thousands to millions of condensates within minutes” specifically refers to conventional flow cytometry, whereas imaging flow cytometry operates at lower throughput (see p. 11). We have also included representative gating strategies for our analysis (see SI p. 26) to ensure clarity and reproducibility. In addition, all raw FCS data have been deposited in Zenodo and are publicly available under DOI: 10.5281/zenodo.17279422, as indicated in the Data Processing and Statistical Analyses section.

We thank the reviewers for their careful evaluation of this manuscript and attention to detail. We feel that the suggested revisions have helped us craft an improved manuscript that will better serve the community.

Sincerely,

Marcos Pires
Professor of Chemistry
Department of Chemistry
University of Virginia
Charlottesville, VA 22904
Email: mpires@virginia.edu

REVIEWERS' COMMENTS, Final Revision_ NCOMMS-25-42050A

Reviewer #1 (Remarks to the Author):

The authors responded well to the 3 reviewers' comments. I agree that the paper is accepted.

RESPONSE: We thank the reviewer for their time and thoughtful consideration of our revisions.

Reviewer #2 (Remarks to the Author):

In the revised version of the manuscript “A High-Throughput, Flow Cytometry Approach to Measure Phase Behavior and Exchange in Biomolecular Condensates”, He and colleagues address many of my comments. Further, their answers should — together with my original comments — be sufficient for the reader to form their own opinion about the robustness of the work.

RESPONSE: We thank the reviewer for their time and thoughtful consideration of our revisions.

Reviewer #3 (Remarks to the Author):

The authors have substantively addressed prior concerns by tempering novelty claims, clarifying the complementary roles of microscopy and FCM, improving figure clarity (Fig. 2 and Fig. 6), and enhancing transparency (gating; data deposition). The study will deliver a robust, scalable platform for population-level condensate kinetics. Accept after the following minor revision:

1. How much “real-time” it is. can authors report the effective dead time from mixing to the first FCM acquisition (e.g., mixing, transfer, start of read). In Fig. 6c–e (pp. 47–48), the earliest point already shows >40% double-positive events, implying that sub-minute exchange may be under-sampled.

RESPONSE: We thank the reviewer for their time and thoughtful consideration of our revisions, and we are sorry for making the omission.

For effective dead time (Fig. 6), a line was added on the figure caption for Fig. 6c to read, " Effective dead time from mixing to the first data acquisition is about 10-

15 seconds; this dead time is not included in the times shown." There is also a change in the caption for Fig. 6e; it reads, "The 10-15 seconds effective dead time is not included in the times shown." A similar line was added in the figure caption for Fig. S11 to read, "Effective dead time from mixing to the first data acquisition is about 10-15 seconds; this dead time is not included in the times shown".

Further, in the Main Text on page 21, text was added to highlight the dead time as follows, "We note that the earliest timepoint reflects an operational dead time of approximately 10–15 seconds between addition of NPM1-Halo-Coumarin, rapid mixing, tube loading, and the initiation of flow-cytometry acquisition. All kinetic assays in this study use identical rapid-handling procedures, ensuring internal consistency across all experiments. Because this mixing dead time already captures a portion of the rapid NPM1 exchange process, the earliest measurement represents droplets that have begun to equilibrate."

2. For multicolor/drug assays in Fig. 5e–f, specify the filters/compensation matrix (briefly is enough) so readers can assess potential color leaks and reproduce the settings.

RESPONSE: We thank the reviewer for their time and thoughtful consideration of our revisions, and we are sorry for making the omission.

For confocal acquisition of drug colocalization images (Fig. 5e), samples were imaged on a Zeiss LSM 980 using 488-nm excitation with a 505–550-nm emission window for FLTX1 (green channel), and 561-nm excitation with a 590–630-nm emission window for mitoxantrone (red channel). These non-overlapping spectral windows minimized bleed-through, and single-color controls confirmed negligible cross-excitation and emission crosstalk.

For flow-cytometry acquisition of drug partitioning assays (Fig. 5f), FLTX1 fluorescence (green channel) was excited with the 488-nm laser and collected in the BL2 detector (525/50-nm bandpass), while mitoxantrone fluorescence (red channel) was excited with the 561-nm laser and collected in the YL2 detector (610/20-nm bandpass). Single-color controls were used to generate a compensation matrix, enabling correction of minor spectral spillover between the green and red channels.

3. In Fig. 6e, state replicate number (n) and error-bar definition (mean \pm SD or SEM) in the caption or Methods?

RESPONSE: We thank the reviewers for their time and thoughtful consideration of our revisions, and we are sorry for making the omission.

In the caption for Fig. 6e, a line has been added to indicate the replicates and error bars as follows, "Each kinetic trace represents a single continuous time-course experiment ($n = 1$) for the respective aging condition. Each time point reflects 10,000 condensate events collected by flow cytometry; no error bars are shown."